# Trends and Uncertainties of Mass-driven Sea-level Change in the Satellite Altimetry Era

Carolina M.L. Camargo[1,2], Riccardo E.M. Riva[2], Tim H.J. Hermans[1,2], and Aimée B.A. Slangen[1]

[1]NIOZ Royal Netherlands Institute for Sea Research, Department of Estuarine and Delta Systems, Yerseke, the Netherlands
[2]Delft University of Technology, Department of Geoscience and Remote Sensing, Delft, the Netherlands

**Correspondence:** Carolina M.L. Camargo (carolina.camargo@nioz.nl)

**Abstract.**

Ocean mass change is one of the main drivers of present-day sea-level change (SLC). Also known as barystatic SLC, ocean mass change is caused by the exchange of freshwater between the land and the ocean, such as melting of continental ice from glaciers and ice sheets, and variations in land water storage. While many studies have quantified the present-day barystatic contribution to global mean SLC, fewer works have looked into regional changes. This study provides an analysis of regional patterns of contemporary mass redistribution associated with barystatic SLC since 1993 (the satellite altimetry era), with a focus on the uncertainty budget. We consider three types of uncertainties: intrinsic (the uncertainty from the data/model itself); temporal (related to the temporal variability in the time series); and spatial-structural (related to the spatial distribution of the mass change sources). Regional patterns (fingerprints) of barystatic SLC are computed from a range of estimates of the individual freshwater sources, and used to analyse the different types of uncertainty. Combining all contributions, we find that regional sea-level trends range from $-0.4$ to $3.3$ $\mathrm{mm.year^{-1}}$ for 2003-2016, and from $-0.3$ to $2.6$ $\mathrm{mm.year^{-1}}$ for 1993-2016, considering the 5-95th percentile range across all grid points and depending on the choice of dataset. When all types of uncertainties from all contributions are combined, the total barystatic uncertainties regionally range from $0.6$ to $1.3$ $\mathrm{mm.year^{-1}}$ for 2003-2016, and from $0.4$ to $0.8$ $\mathrm{mm.year^{-1}}$ for 1993-2016, also depending on the dataset choice. We find that the temporal uncertainty dominates the budget, responsible on average for $65\%$ of the total uncertainty, followed by the spatial-structural and intrinsic uncertainties, which contribute on average $16\%$ and $18\%$, respectively. The main source of uncertainty is the temporal uncertainty from the land water storage contribution, which is responsible for $35-60\%$ of the total uncertainty, depending on the region of interest. Another important contribution comes from the spatial-structural uncertainty from Antarctica and land water storage, which shows that different locations of mass change can lead to trend deviations larger than $20\%$. As the barystatic SLC contribution and its uncertainty vary significantly from region to region, better insights into regional SLC are important for local management and adaptation planning.

**Plain Language Summary**

The mass loss from Antarctica, Greenland and glaciers, and variations in land water storage cause sea-level changes. Here, we characterise the regional trends within these sea level contributions, taking into account mass variations since 1993. We take a

comprehensive approach to determining the uncertainties of these sea-level changes, considering different types of errors. Our study reveals the importance of clearly quantifying the uncertainties of sea-level change trends.

**Keywords**

Ocean Mass; Sea-level change; Sea-level equation; Ice sheets; Glaciers; Land Water Storage; Uncertainties

## 1 Introduction

Even if all countries respect to the Paris Agreement, global mean sea level will continue to rise in the coming decades and beyond (Wigley, 2005; Nicholls et al., 2007; Oppenheimer et al., 2019; Fox-Kemper et al., 2021). The reason for this is the long response time of the ocean and the cryosphere to climate change (Abram et al., 2019). As a consequence, coastal societies all over the world will need to deal with a certain amount of sea-level change (SLC). Therefore, a good understanding of present-day SLC and its drivers is required, as it yields better future sea-level projections, which are necessary for adaption and mitigation planning.

The attribution of SLC to its different drivers is known as the sea-level budget (WCRP, 2018). Alongside density driven (steric) changes (e.g., MacIntosh et al. (2017); Camargo et al. (2020)), present-day SLC is mainly driven by the mass loss of continental ice stored in glaciers and ice sheets, and by variations in land water storage (LWS) (WCRP, 2018; Fox-Kemper et al., 2021). The contribution of ocean mass changes, termed barystatic SLC (Gregory et al., 2019), was responsible for about 60% of the global mean SLC over the 20th century (Frederikse et al., 2020; Fox-Kemper et al., 2021). Barystatic SLC varies significantly from region to region and strongly depends on the location of terrestrial mass loss (Mitrovica et al., 2001). For example, a collapse of the West Antarctic Ice Sheet would cause sea level to rise 1.6 times more in San Francisco (US) than in Santiago (Chile) (Gomez et al., 2010). Thus, for local management and climate planning, it is important to understand the barystatic contribution to regional SLC (Larour et al., 2017).

The regional patterns associated with barystatic SLC can be computed by solving the sea-level equation (SLE) (Farrell and Clark, 1976), which results in the so-called sea-level fingerprints (Mitrovica et al., 2001). These patterns reflect the so-called gravitational, rotational and deformation (GRD) response of the Earth to mass redistribution (Gregory et al., 2019). GRD-induced sea-level fingerprints have been the subject of several studies, ranging in scope from paleoclimatic SLC, for example due to the last deglaciation event (Lin et al., 2021), to contemporary SLC (Frederikse et al., 2020) and future sea-level projections (e.g., Slangen et al. (2012, 2014)). Most of the studies including present-day mass contributions have focused either on the GRACE satellite period (since 2002) (Bamber and Riva, 2010; Riva et al., 2010; Hsu and Velicogna, 2017; Adhikari et al., 2019; Frederikse et al., 2019), on the closure of the sea-level budget over a longer period (Slangen et al., 2014; Frederikse et al., 2020) or on their contribution to global mean SLC (Chambers et al., 2007; Horwath et al., 2021). However, an in-depth analysis of the GRD-induced regional patterns associated with barystatic SLC and its uncertainties during the satellite altimetry era (since 1993) has not yet been done. Insights into the contemporary contributions of ice sheets, glaciers and land

water storage to regional SLC and their uncertainties over the last three decades are important to constrain regional sea-level projections and obtain a better closure of the regional sea-level budget.

The importance of quantifying the uncertainties in sea-level studies has increasingly received attention (Bos et al., 2014; Royston et al., 2018; Ablain et al., 2019; Camargo et al., 2020; Palmer et al., 2021; Prandi et al., 2021; Horwath et al., 2021). One of the approaches to describe the uncertainties of a system is to partition the total uncertainty budget into different kinds of uncertainties. Errors in the measurement system, known as intrinsic uncertainties (Palmer et al., 2021), describe the sensitivities of choices within a methodology (Thorne, 2021). The intrinsic uncertainties, also referred as observational (Ablain et al., 2019; Prandi et al., 2021) or parametric (Thorne, 2021), need to be determined during the low-level data processing and are usually provided with higher level (ready-to-use) products. Another class of uncertainties originates from the use of different methodologies to describe the same physical system, known as structural uncertainty (Thorne et al., 2005; Palmer et al., 2021). This can be defined as the spread around a central (ensemble) estimate. The structural uncertainty is related to the use of different datasets of the same process. Note that, if different datasets use the same product for corrections, calibrations and/or validation, the intrinsic and structural uncertainties could be partially correlated. Regarding the GRD-induced pattern associated with barystatic SLC, the spread in the location of the mass change introduces another source of error, which we call spatial uncertainty. Finally, another type of uncertainty results from the autocorrelation of the observations (Bos et al., 2013), which we refer to as temporal uncertainty. This uncertainty becomes relevant when a functional model, such as a (linear) trend, is used to describe the changes within the system. The temporal uncertainty can be estimated by using noise models while determining the trend. Together, the intrinsic, structural, spatial and temporal uncertainties describe the uncertainties of an observed quantity, in this case the GRD-induced pattern associated with barystatic SLC.

The aim of this work is to provide a comprehensive overview of barystatic SLC and the associated regional GRD-induced patterns with a focus on the global and regional uncertainty budget. Throughout this paper, we use 'GRD-induced SLC' when referring to the GRD-induced regional pattern associated with barystatic SLC. We use state-of-the-art datasets of mass contributions from land ice and LWS (Section 2.1) to compute regional sea-level fingerprints (Section 2.2.1). In addition, we present a methodological framework to describe the uncertainties of the fingerprints (Section 2.2.2). We follow the noise model analysis of Camargo et al. (2020) to quantify the *temporal uncertainty* (Section 3.1;3.2). We combine the effect of ice geometry on sea-level fingerprints (Bamber and Riva, 2010; Mitrovica et al., 2011) with the structural uncertainty definition of Palmer et al. (2021), to compute the *spatial-structural uncertainty* of the fingerprints (Section 3.3). Together with the *intrinsic uncertainty* (Section 3.4), we present the total GRD-induced SLC trend and uncertainty for 2003-2016 and 1993-2016 (Section 3.5). We finalize this manuscript with an overview and a discussion of our findings (Section 4).

## 2  Data and Methodology

### 2.1  Datasets

To obtain the GRD-induced SLC patterns we use a range of estimates of mass changes of the Antarctic and Greenland ice sheets (AIS and GIS, respectively), glaciers (GLA), and land water storage (LWS) . We define LWS anomalies as water mass

changes outside glacierized areas: the sum of water stored in rivers, lakes, wetlands, artificial reservoirs, snow pack, canopy
and soil (groundwater) (Cáceres et al., 2020). For each of the contributions we use four different estimates (Table 1, Figure
1, and discussed in more detail in Supplementary Text A). Despite the methodological differences between the datasets, they
show a good agreement in reproducing the global mean barystatic sea-level changes (Figure 1)

**Table 1.** Overview of datasets used in this manuscript.

| Contribution | Dataset | Temporal range | Source | Dependence* | Acronym | Spatial Resolution |
|---|---|---|---|---|---|---|
| All | CSR mascon RL06 | 2003-2020 | observations | GRACE(-FO) | CSR | $1°$x$1°$ ** |
| | JPL mascon RL06 | 2003-2020 | observations | GRACE(-FO) | JPL | $3°$x$3°$ ** |
| AIS | IMBIE 2018 | 1993-2016 | ensemble datasets | Hybrid | IMB | Region mean |
| | Rignot 2019 | 1979-2017 | observations + model | Independent | UCI | Drainage basin mean |
| GIS | IMBIE 2020 | 1993-2018 | ensemble datasets | Hybrid | IMB | Region mean |
| | Mouginot 2019 | 1972-2018 | observations + model | Independent | UCI | Drainage basin mean |
| Glaciers | Zemp 2019 | 1962-2016 | observations + model | Independent | ZMP | Glacier mean |
| | WaterGAP | 1958-2016 | glaciers model | Independent | WGP | $0.5°$ |
| LWS | WaterGAP | 1958-2016 | hydrological model | Independent | WGP | $0.5°$ |
| | PCR-GLOBWB | 1948-2016 | hydrological model | Independent | GWB | $5 arcmin$ |

*Dataset dependence on GRACE; **Note that while the mascons are provided in $0.25°$ and $0.5°$ resolution, the native resolution of the mascons solution are $1°$x$1°$ and $3°$x$3°$
equal-area grids at the equator for CSR and JPL, respectively (Save et al., 2016; Watkins et al., 2015).

One of the main sources of observations of Earth's mass changes is the satellite mission Gravity Recovery and Climate
Experiment (GRACE, Tapley et al. (2004)) and its follow-on mission (GRACE-FO, Landerer et al. (2020)). We use GRACE
mass concentrations (mascons) over land as estimates of changes in AIS, GIS, glaciers and LWS. To avoid methodological
biases, we use mascon solutions from two different processing centres: RL06 from Center for Spatial Resarch (CSR) (Save
et al., 2016; Save, 2020) and RL06 v02 from Jet Propulsion Laboratory (JPL) (Watkins et al., 2015; Wiese et al., 2019) (Table
1). JPL and CSR mascons are provided on a $0.5°$ and $0.25°$ lon-lat grid, respectively, but they actually are resampled from the
native $3°$x$3°$ and $1°$x$1°$ equal-area grids (Save et al., 2016; Watkins et al., 2015). Considering the native resolution of GRACE
observations of about 300km at the equator (Tapley et al., 2004), the JPL mascons should have independent solutions at each
mascon centres, with uncorrelated errors, while the CSR mascons are not fully independent of each other and are expected to
contain spatially correlated errors.

To isolate the individual contributions of AIS, GIS, LWS and GLA in the GRACE mascons, we use an ocean-land-cryosphere
mask (Supplementary figure A1), which delineates the drainage basins of the ice sheets (based on Mouginot and Rignot (2019),
Rignot et al. (2011)), the glaciers (based on the Randolph Glacier Inventory, Consortium (2017)), and the remaining land
regions (based on ETOPO1, Amante and Eakins (2009)). Considering the size of glaciers, the resolution of the GRACE signal
is not high enough to (i) separate the peripheral glaciers from the ice sheets, and (ii) to separate the signal of glaciers and

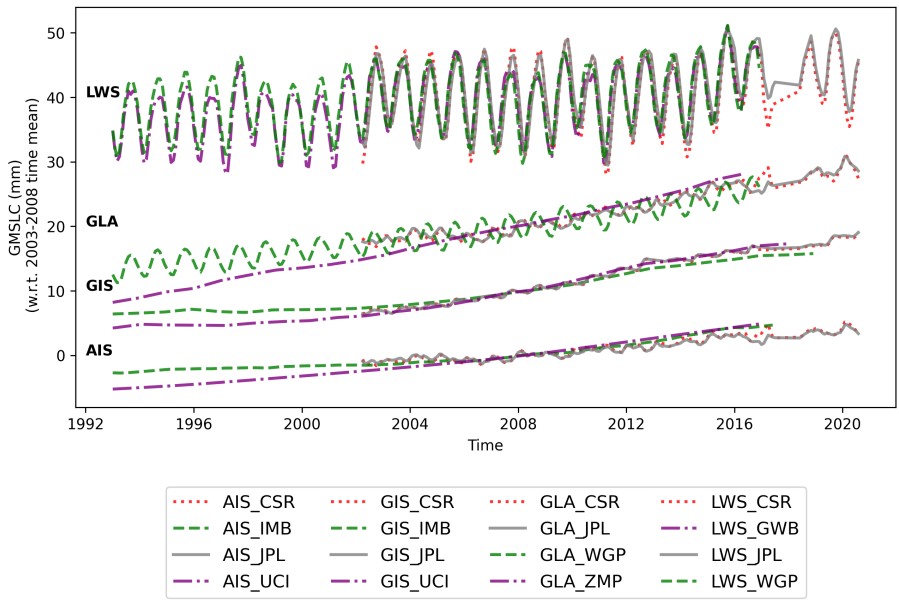

**Figure 1.** Global mean barystatic sea-level change time series. Different components are vertically offset for visualization purposes.

LWS in regions with small glacier coverage and large LWS contribution. Thus, to isolate the glaciers signals from the mascons we follow the method described in Reager et al. (2016) and Frederikse et al. (2019): (1) peripheral glaciers to Greenland and
Antarctica are included with the ice sheets mass changes; (2) regions where glaciers dominate the mass changes are considered 'full' glaciers, that is, the land signals in those regions are purely denoted as glacier mass change. These include the RGI regions of Alaska, Arctic Canada North, Arctic Canada South, Iceland, Svalbard, Russian Arctic Islands and Southern Andes; (3) for the remaining glaciated regions, we assume that the mass change is partly due to glacier mass change, and partly due to LWS ('split' glaciers). In these regions the glacier mass changes are known to be small and mass changes are dominated by
LWS. We use the glacier estimates of Hugonnet et al. (2021), which are based on satellite and airborne elevation datasets as our glacier estimates in these regions. Unlike gravimetry observations, the estimates of Hugonnet et al. (2021) do not include the hydrological 'contamination'. To isolate the glacier from the LWS signal, we subtract the corrected glacier estimates from the total mass change in the mascons. The remaining signal is then added to the LWS contribution.

     Apart from GRACE data, which is only available since late 2002, we use seven other datasets in our analysis, from which
five are independent of GRACE and two partly incorporate GRACE information (Table 1. For LWS, we use data from two global hydrological models: PCR-GLOBWB (GWB, Sutanudjaja et al. (2018)) and WaterGAP (WGP, Cáceres et al. (2020)). The latter also incorporates a time series of glacier mass variations from the global glacier model of Marzeion et al. (2012). We use the ocean-land-cryosphere mask (Supplementary figure A1) to separate the LWS and GLA estimated from WGP. For GLA, in addition to the WGP model simulations, we also use observational estimates from Zemp et al. (2019), which are based
on an extrapolation of glaciological and geodetic observations. For the GIS and AIS, we use observation- and model-based

data from Mouginot et al. (2019) and Rignot et al. (2019), respectively. We refer to these as UCI datasets, since they were both developed at the University of California at Irvine (UCI). We also use AIS and GIS estimates from the ice sheet mass balance inter-comparison exercise (IMBIE, Shepherd et al. (2018, 2020)), which combines ice sheet mass balance estimates developed from three different techniques (satellite altimetry, satellite gravimetry (GRACE) and the input-output method).

## 2.2 Methodological Framework

We characterize GRD-induced SLC by a linear trend and the three types of uncertainties discussed earlier. We use the following time periods for the trend analysis: from 1993-2016 for the non-GRACE datasets, and from 2003-2016 for all datasets. The framework used to compute and combine the uncertainties and associated regional sea-level patterns is schematized in Figure 2. The main modules of the framework (bold text in the blue boxes of Figure 2a) are further explained in Figure 2b and in sections 2.2.1 and 2.2.2.

The trends and associated temporal uncertainties are estimated directly from the mass source time series (Table 1) in the *noise model* module (Figure 2a). Thus the noise model analysis (Section 3.1) describes the physical processes of the mass sources instead of the temporal correlation in the sea-level fingerprint. The mass source change trend and temporal uncertainty are then used as input to the *SLE model* module (Section 2.2.1), which computes how the mass changes on land affect regional ocean mass change (i.e., GRD-induced SLC; Section 3.2). The mass source trends are also used as input to the *spatial uncertainty* analysis (Section 3.3). The uncertainty of the mass source time series is used as input to the *intrinsic uncertainty* analysis (Section 3.4).

### 2.2.1 The Sea-Level Equation Model

The regional GRD-induced SLC patterns resulting from the barystatic contributions can be computed by solving the sea-level equation (SLE) (Farrell and Clark, 1976), using spatial and temporal information of GLA, AIS, GIS and LWS (Mitrovica et al., 2001; Tamisiea and Mitrovica, 2011). Before computing the regional SLC fields, all input data (Table 1) is converted to equivalent water height, and bilinearly interpolated to a 1° by 1° grid. The SLE model then computes how the source mass change is redistributed over the oceans, taking into account the GRD response of the Earth to these mass changes (Milne and Mitrovica, 1998; Mitrovica et al., 2001; Tamisiea and Mitrovica, 2011). The SLE model uses a pseudospectral approach (Mitrovica and Peltier, 1991) up to spherical harmonic degree and order 180 (equivalent to a spatial resolution of one degree). We assume a purely elastic solid-Earth response to the mass redistribution, based on the Preliminary Reference Earth Model (Dziewonski and Anderson, 1981). While we focus here on the fingerprints of relative SLC, that is, the difference in height between the geoid and the solid Earth surface, we also provide the complementary geocentric (absolute) fingerprints (see *Data availability* Section).

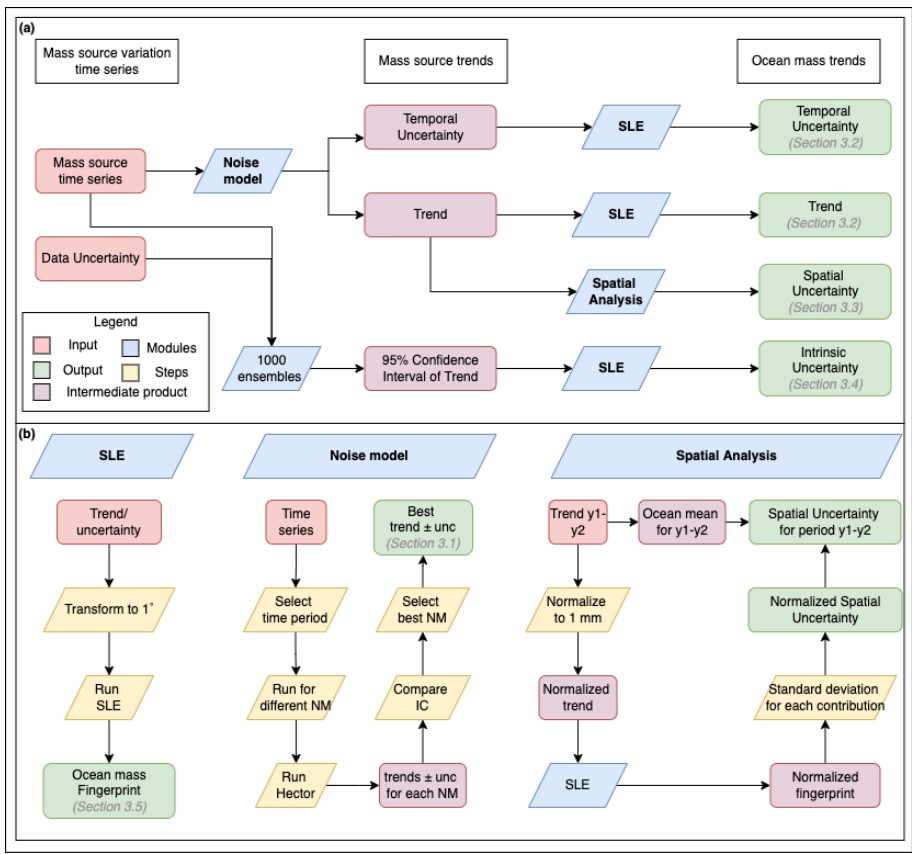

**Figure 2.** Overview of the framework used in this study (a), with detailed modules (b). Red boxes indicate the initial data (Table 1), purple the intermediate products, and green the final products. The yellow boxes indicate steps of the methodology, and the blue the main modules. We use the following acronyms and abbreviations: OLS: ordinary least-squares; SLE; Sea-level equation; IC: Information Criteria; unc: uncertainty; NM: noise model; Hector: software package by Bos et al. (2013).

## 2.2.2 Trend and Uncertainty Assessment

Our GRD-induced SLC and associated temporal uncertainty (Figure 2, centre column) are computed using the software package Hector (Bos et al., 2013), in which the observations are assumed to be the sum of a deterministic model (including annual and semi-annual signals) and stochastic noise. Different noise models can be selected to describe the autocorrelation between the residuals of the regression. The uncertainty of the regression model, representing one standard deviation, is then used as our temporal uncertainty.

Based on previous studies (Bos et al., 2013; Royston et al., 2018; Camargo et al., 2020), we test eight noise models to find the best descriptor of the uncertainties in our data:

– white noise (WN), in which no autocorrelation between the residuals is considered;

- pure power law (PL), where all observations influence one another, although their correlation decreases with increasing temporal distance;

- PL combined with WN (PLWN);

- auto-regressive of orders 1, 5, and 9 (AR(1), AR(5), and AR(9), respectively), in which the order represents the number of previous observations influencing the next one;

- autoregressive fractionally integrated moving average of order 1 (ARF), which combines an AR(1) model with a fractional integration and a moving average of the noise;

- generalized Gauss-Markov (GGM), a generalized form of the ARF model.

The goodness of the fit of the models is assessed with the modified Bayesian Information Criterion ($BIC_{tp}$; He et al. (2019)), which is an intermediate criterion in relation to the Akaike (AIC; Akaike (1974)) and Bayesian (BIC; Schwarz (1978)) criteria. The best noise model is the one that minimizes these criteria. Since these criteria are relative values, they can not be compared between different data sets. Thus, we compare the criteria of different noise models for each data set and each grid point separately. To select the best noise model, we compute the relative likelihood of the $BIC_{tp}$, and select the model with values smaller than 2 (Burnham and Anderson, 2002; Camargo et al., 2020). Note that all noise models reasonably capture the variability of the time series (Figure A2), as their scores are always within a similar range.

The second uncertainty we consider is the spatial-structural uncertainty (Figure 1b, right column). Studies that combine a large number of datasets often base the structural uncertainty of an estimate on the standard deviation over the individual datasets in relation to the ensemble mean (Palmer et al., 2021; Cazenave et al., 2018). To isolate the effect that the spatial distribution of the terrestrial mass change has on the fingerprints, we compute the spatial-structural uncertainty by estimating the standard deviation for each contribution based on normalized fingerprints. The latter means that the sum of the regional SLC for each contribution is equal to 1 $\mathrm{mm.year^{-1}}$ of SLC. By using normalized fingerprints we remove the weight that the different central estimates (mean) have on the spatial standard deviation. We then take the standard deviation across the four normalized datasets for each mass source contribution, obtaining four normalized spatial-structural uncertainties, which reflects the uncertainty associated with the different spatial resolutions and location of mass change of the datasets. For example, the spatial-structural uncertainty of the AIS reflects the differences in the fingerprints due to the fact that GRACE datasets provide observations at a 0.25 degrees resolution, while UCI provides mass changes averaged over the 17 main drainage basins of the ice sheet, and IMBIE mass changes averaged over three regions of the ice sheet (west, east and peninsula). While the analysis is based on the 2003-2016 trend, we assume that the normalized fingerprints are time-invariant, and that the resulting uncertainty is also representative of the 1993-2016 period. Lastly, we multiply the normalized uncertainty by the ocean mean (central estimate) of each contribution for 1993-2016 and 2003-2016 to compute the spatial-structural uncertainty for the respective period. We note that all components show some decadal variability in the spatial distribution, and thus assuming that the spatial mass change distributions from 2003-2016 are representative of the period 1993-2016 is an approximation of the study. However, by multiplying the normalized fingerprint by the mean of each period the possible error from this assumption

becomes fairly limited. Furthermore, using a shorter spatially dense time series to obtain the variability of a longer period when only limited information is available is a methodology that is often used in sea-level studies (e.g., Church and White (2006); Frederikse et al. (2020)).

The final type of uncertainty considered in our assessment is the intrinsic uncertainty, which represents the formal errors and sensitivities in the measurement system and needs to be provided with the observations/models by the data processor/distribution centre. The intrinsic uncertainty was only provided with the JPL and IMBIE datasets. For all other datasets, our uncertainty budget does not include the intrinsic uncertainty. The uncertainties provided with the JPL Mascons represent the scaling and leakage errors from the mascon approach (Wiese et al., 2016), and, over land, are scaled to roughly match the

formal GRACE uncertainty of Wahr et al. (2006). The latter represent errors in monthly GRACE gravity solutions, encompassing measurement, processing and aliasing errors (Wahr et al., 2006). While the mascons have been corrected for mass changes due to glacial isostatic adjustment (GIA) with the ICE6G-D model (Peltier et al., 2018), the intrinsic uncertainties of the JPL mascons do not represent the uncertainties from the GIA correction, which can be large depending on the region (Reager et al., 2016; Wouters et al., 2019). For example, the choice of the GIA model used for the correction could lead to

uncertainties representing up to $19\%$ of the signal in Antarctica, but less than $1\%$ in Greenland (Blazquez et al., 2018). Given that estimating GIA uncertainties is in itself an open issue (Caron et al., 2018; Simon and Riva, 2020), we could not propagate full GIA uncertainties into the fingerprints. Since the intrinsic uncertainty represents systematic errors and instrumental noise, which might be serially correlated, we assume that the errors can be approximated by a random walk. We therefore generate an ensemble of 1,000 time series by perturbing the original rate with random normal noise multiplied by the uncertainty time

series. We then compute the trend for each ensemble member. We use half of the width of the $95\%$ CI as input in the SLE model to show how the mass associated with the intrinsic uncertainty is distributed over the oceans.

### 2.2.3   Combining Trends and Uncertainties

To compute total GRD-induced SLC trends and their uncertainties, we sum the individual contributions (AIS, GIS, LWS and GLA) as follows, with a total of six combinations: 1.CSR (all); 2.JPL (all); 3.IMB (AIS/GIS) + WGP (LWS/GLA); 4.UCI

(AIS/GIS) + WGP (LWS/GLA); 5.IMB (AIS/GIS) + GWB (LWS) + ZMP (GLA); and 6.UCI (AIS/GIS) + GWB (LWS) + ZMP (GLA).

    Whereas the trends are added together linearly, we add the uncertainties in quadrature, assuming they are independent and normally distributed. We acknowledge that this is an important assumption, as it is possible that the intrinsic uncertainty will be reflected in the temporal and structural uncertainties. However, we keep the independence assumption to obtain a more realistic

(and smaller) estimate of the final uncertainty (Taylor, 1997). For each contribution, we first combine the different types of uncertainty following Equation (1):

$$\sigma_{CONTR} = \sqrt{\sigma_{temporal}^2 + \sigma_{spatial}^2 + \sigma_{intrinsic}^2} \qquad (1)$$

where $\sigma_{CONTR}$ is the total uncertainty for each individual contribution (AIS, GIS, GLA, LWS). We then compute the total GRD-induced uncertainty for all contributions ($\sigma_{total}$) following Equation (2):

$$\sigma_{total} = \sqrt{\sigma_{AIS}^2 + \sigma_{GIS}^2 + \sigma_{LWS}^2 + \sigma_{GLA}^2} \qquad (2)$$

## 3 Results

In this Section we first present the noise model selection (Section 3.1) used to compute the GRD-induced SLC trend and temporal uncertainty (Section 3.2). We then present the spatial-structural (Section 3.3) and intrinsic uncertainties (Section 3.4). Lastly, we show the total GRD-induced SLC trends (i.e., the sum of the different contributions) and uncertainties (i.e., the sum of the different contributions and types of uncertainties) and zoom in on a few coastal examples (Section 3.5).

### 3.1 Noise characteristics of the mass sources

Many geophysical time-series are known to exhibit temporal (auto)correlations, as is the case for sea-level and cryosphere data (Bos et al., 2013). This autocorrelation means that each observation is not completely independent from the previous one (Bos et al., 2013), and it is defined by the shape of the spectrum of the time-series (Hughes and Williams, 2010). Understanding the shape of spectra and determining the best stochastic model to describe these spectra is important to understand the physics of the processes playing a role in the time-series (Hughes and Williams, 2010). In addition, accounting for the autocorrelation of the time-series while estimating a linear trend is important both for the value of the trend itself and for the statistical error of the fit (Bos et al., 2013; Hughes and Williams, 2010). Depending on the nature of the process being studied, different noise models can be used to account for the effects of autocorrelations. Here, we determine the best noise model for each spatial data point of the mass sources of the different barystatic contributions (AIS, GIS, LWS, GLA). Our analysis shows that the optimal noise model depends on both the physical system (AIS, GIS, GLA or LWS) and the dataset (Figure 3).

There are clear differences between the GRACE datasets (Figure 3a-h), for which the PL and GGM noise models score higher, and the other datasets (Figure 3i-p), for which the AR(5) and AR(9) models score higher. The only exception is for the two Greenland datasets (GIS_JPL (f) and GIS_IMB (j)), where the noise model selection is reversed. Over the ice sheets, the higher resolution of GRACE observations (compared to IMBIE and UCI datasets) leads to more heterogeneity in the model selection, which suggests the inclusion/capture of more complex processes. For example, our analysis indicates that only one type of noise model is selected for the entire ice sheet in the IMBIE dataset (Figure 3i-j). For LWS changes, where the spatial resolution of GRACE and the hydrological models is relatively high, the noise model selection follows a different pattern. There is a general preference for AR(1) in areas with smaller LWS changes (i.e., not the large drainage basins). On the other hand, over the large drainage basins, the same model preference mentioned above is maintained (Figure 3, right column). This suggests that GRACE observations and the hydrological models might not always be capturing the same processes.

Different noise models are selected as optimal for the two GRACE datasets: CSR datasets (Figure 3a-d) are best explained with the PL model, while JPL estimates (Figure 3e-h) are best explained with the GGM model. However, the GGM model is

fairly similar to a pure power-law model under certain parameters. Furthermore, the noise model selection for the CSR dataset

over the ice sheets (Figure 3a,b) displays an interesting pattern, which is not seen for the JPL dataset (Figure 3e,f). Regions with relatively strong ice melt (i.e., the Antarctica Peninsula, East Antarctica and northwest of Greenland) are better represented by an AR(5) model. Over the extremities of the ice sheets, which are more dynamic regions, the GGM model is the optimal one. On the other hand, internal regions of the ice sheets, where there is little ablation, are better described by the PL model.

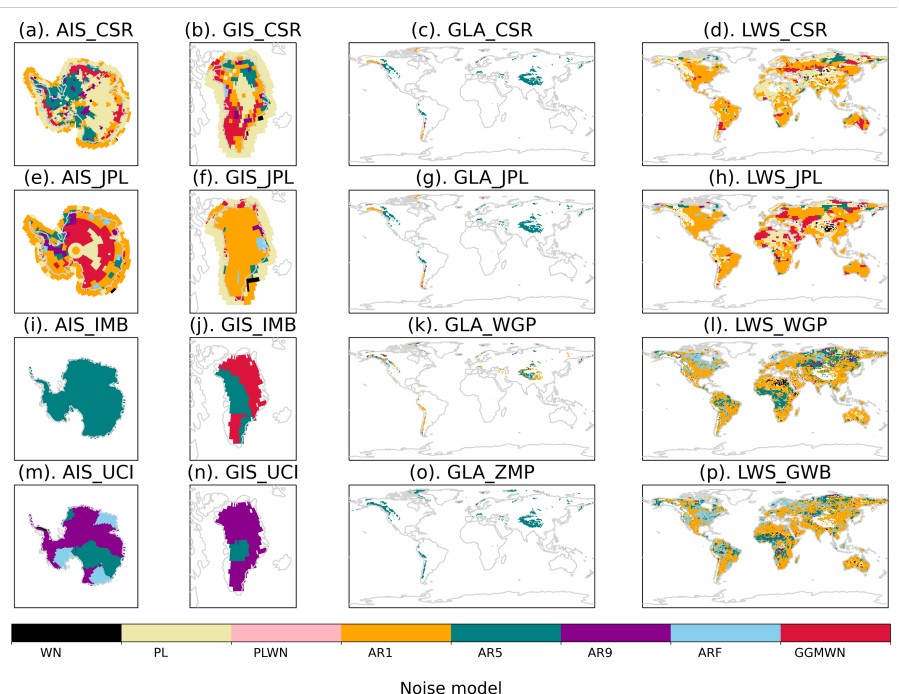

**Figure 3.** Noise model selection based on the time series of the different sources of mass loss for each dataset (rows) and contribution (columns), over the period 2003-2016.

## 3.2 Trend and temporal uncertainty

The mass source trend and uncertainties obtained with the selected noise models (Section 3.1) are used to compute the sea-level fingerprints (Figure 4). To illustrate the difference between the fingerprints based on GRACE and those based on GRACE-independent datasets, we show the trends and uncertainties for the JPL estimates (Figure 4a-d, i-l) and for the UCI estimates for the ice sheets (Figure 4e-h) and WaterGAP for glaciers and LWS (Figure 4m-p). Trends and temporal uncertainties for the other datasets are provided in Figure A3. The typical GRD patterns are visible in all fingerprints: regions closer to a freshwater

source experience a negative SLC, due to the mass loss that causes land uplift and reduced gravitational attraction, while in the far-field the sea level rises more than the global average.

While all trends strongly depend on the dataset (Figure 4, first and third column), the uncertainty patterns are rather consistent. This suggests that, even though different noise models were used to compute the trend for each dataset, the temporal uncertainty is characteristic of each contribution. We find that for any given contribution, the trends from different datasets are consistent within their respective uncertainties. For glaciers and the ice sheets, the GRACE-independent datasets give a higher trend than the GRACE observations. The temporal uncertainties for ice sheets and glaciers are relatively small, especially for the UCI datasets. This indicates that these contributions do not exhibit strong autocorrelations, and consequently the uncertainty of the trend will be small. On the other hand, the temporal uncertainty for the LWS is larger than the trend itself, and therefore the LWS trend is not statistically significant. This is probably related to the large internal and decadal variability of the time series, in combination with the relatively short period under study.

The largest inter-dataset differences are displayed in the regional patterns of the LWS contribution. Despite the similar global mean LWS trend value for both JPL and WGP, the regional trend patterns and uncertainty values are very different. This may partially be related to the coarse resolution of GRACE (300 km) in comparison to the hydrological models ($0.5°$ by $0.5°$ grid (55 km by 55 km at the Equator)). This difference can also be related to the difficulty in modelling the complex processes affecting LWS, which relies on parameterisations of physical processes and on sparse observations, while GRACE measures the total mass change.

Another significant inter-dataset difference is in the regional trend pattern as a consequence of AIS mass change (Figure 4a,e). This is mainly related to the location of ice mass changes in each dataset. GRACE observes mass accumulation in East Antarctica, resulting in a positive sea-level trend in the region. This accumulation is not captured by the UCI and IMB data sets. GRACE has a higher spatial resolution, and thus provides more detail of where the mass change is taking place. The UCI dataset provides estimates on a basin scale, so more detailed changes may be averaged out. The effect of the location of mass change at the source of the contribution is further investigated with the spatial-structural uncertainty (next section).

### 3.3 Spatial-structural uncertainty

The regional SLC fingerprints directly reflect the differences in the spatial distribution of the mass change sources of the datasets (Mitrovica et al., 2011). Over the ice sheets, for instance, IMBIE provides one time series for the entire Greenland Ice Sheet, which is subdivided into dynamic and surface mass balance changes, and the Antarctic Ice Sheet is divided into three drainage basins. GRACE products, on the other hand, have a native resolution of about 300-km at the equator (Tapley et al., 2004). To account for the uncertainties arising from the differences in location of the mass change between datasets, we first normalize the fingerprints and then combine them into estimates of the spatial-structural uncertainty (Figure 5).

For all contributions, the largest spatial uncertainties are concentrated closer to the mass change sources, while the uncertainties are reduced in the far field. The effect of differences resulting from Earth rotational effects (typically leading to four large quadrants) is visible in the far field of the AIS (in the Northern Pacific) and near hotspots of LWS (around the Southern Ocean). As was the case for the trends (Figure 4a), the AIS shows the strongest spatial differences, as the underlying datasets strongly differ in their spatial detail. The spatial uncertainties represent the error introduced by using datasets that have insufficient resolution to solve the processes being analysed. In addition, it also shows that different physical processes are captured

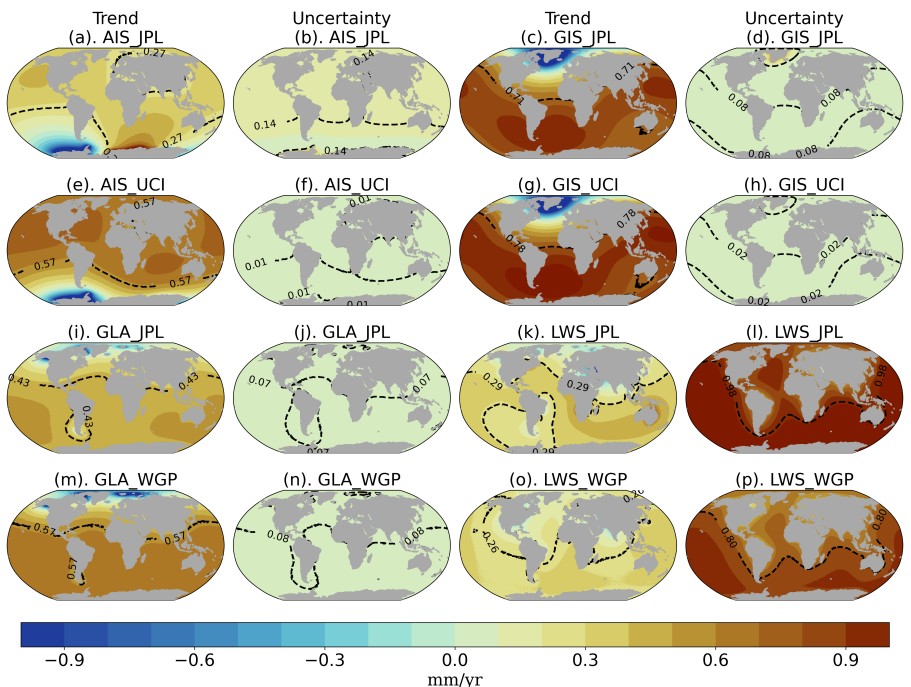

**Figure 4.** GRD-induced sea-level trend and temporal uncertainty $(\mathrm{mm.year}^{-1})$ for GRACE (JPL) and independent combination (UCI + WGP) for 2003-2016. Black dashed contour line and number indicates the spatial average of the regional trend and uncertainty. Trends and uncertainties of CSR, IMB, ZMP and GWB presented in Supplementary Figure A3

by the different datasets, as is the case for the LWS estimate. The discrepancies between the processes captured by GRACE and LWS models result in the spatial-structural uncertainty of the LWS component (Figure 4d) being the second largest.

### 3.4 Intrinsic uncertainty

The final type of uncertainty considered here is the intrinsic uncertainty, which represents noise related to the dataset itself (Figure 6). With exception of the LWS, all intrinsic uncertainties are relatively small (spatial averages below $0.1\mathrm{mm.year}^{-1}$). The largest intrinsic uncertainty is seen in the LWS contribution (Figure 6a), with maximum values of 0.5 $\mathrm{mm.year}^{-1}$. This is expected, as the uncertainty of GRACE is estimated from the standard deviation of the signal anomalies (Wahr et al., 2006), which may lead to an overestimation of the uncertainty in regions where the anomalies represent real hydrological signals (Humphrey and Gudmundsson, 2019). Furthermore, GRACE mass errors are latitude dependent, increasing from the poles to the equator (Wahr et al., 2006), which explains why we see large intrinsic uncertainty for LWS and low values for the ice sheets and glaciers. The IMBIE datasets (Figure 6e,f) show larger intrinsic uncertainty than the ice sheet uncertainties from JPL (Figure 6c,d), once the IMBIE time series is an ensemble of several datasets and methods. Note that these uncertainties are smaller than those originally reported in the IMBIE studies (Shepherd et al., 2018, 2020), which include not only intrinsic,

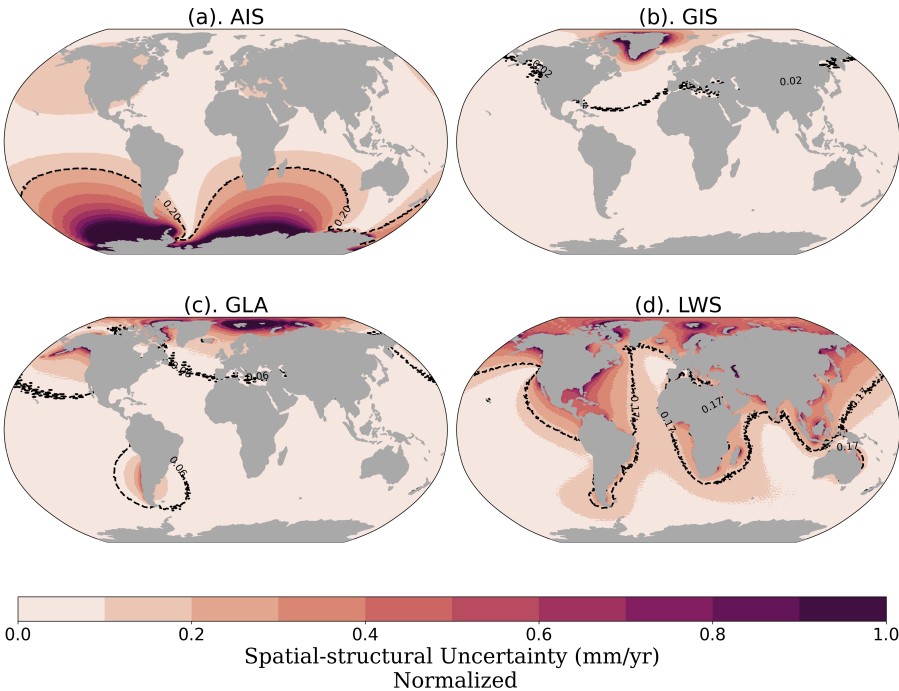

**Figure 5.** Normalised GRD-induced sea-level change fields of the spatial-structural uncertainty (0-1 $\text{mm.year}^{-1}$), representing the uncertainty arising from the different locations of mass changes for Antarctica (a), Greenland (b), glaciers (c) and land water storage (d). Black dashed contour line and number indicates the spatial average of the regional uncertainty.

but also structural and temporal uncertainties. Overall, the intrinsic uncertainty, which depends on the method employed to produce the estimates, is small compared to the spatial-structural and temporal uncertainties, which are related to the physical processes represented.

## 3.5 Total Barystatic Trend and Uncertainty

Combining the different contributions, as explained in Section 2.2.3, leads to the total GRD-induced SLC trends and uncertainties shown in Figure 7. Although we analysed six dataset combinations, here we show only two (JPL and IMB+WGP) to discuss the patterns and the total uncertainty fields. We show these specific combinations because they present the most complete uncertainty budget (as only JPL and IMB provided intrinsic uncertainties). Additional combinations are presented in Supplementary Figure A4, with the global mean barystatic SLC values listed in Supplementary Table A1. We recall that the aim of this study is not to provide one final ensemble of GRD-induced SLC, but rather to focus on the uncertainty budget. Figure 7 shows the JPL GRACE dataset (panels a-b) and the combination of IMBIE and WaterGAP (c-f), the latter for both the common period of 2003-2016 (a-d) and the longer period of 1993-2016 (e-f). To illustrate the distribution of the regional trends and uncertainties around the world, we report the 5th to 95th percentile range across all ocean grid cells (Figure 7,

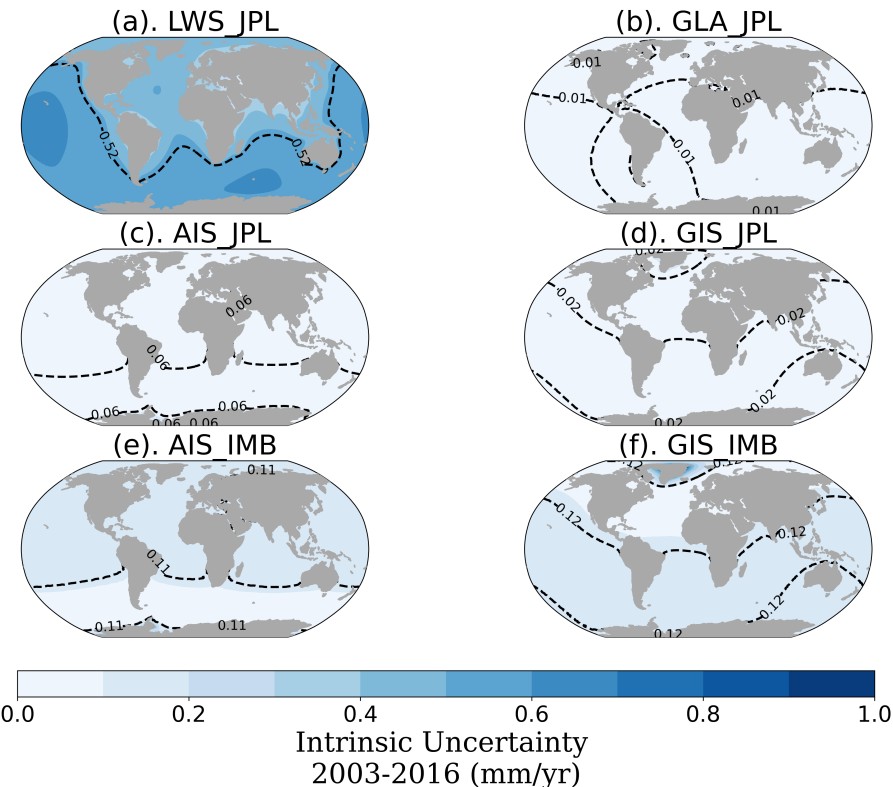

**Figure 6.** GRD-induced sea level fields of the intrinsic uncertainty (mm.year$^{-1}$) for the land water storage (a), glaciers (b), Antarctica (c) and Greenland (d) contributions of the JPL dataset; and Antarctica (e) and Greenland (f) contributions of the IMBIE dataset. Black dashed contour line indicates the spatial average of the regional uncertainty

histograms below the maps), and refer to it as the 90%-range of the field. When all the contributions are combined, we find that the 90%-range of the GRD-induced SLC trends range from $-0.43$ to $3.31$ mm.year$^{-1}$ for 2003-2016, and from $-0.32$ to $2.56$ mm.year$^{-1}$ for 1993-2016, depending on the dataset choice and the location. When all types of uncertainties from all contributions are combined, the 90%-range of GRD-induced total uncertainty ranges from $0.61$ to $1.27$ mm.year$^{-1}$ for 2003-2016, and from $0.36$ to $0.79$ mm.year$^{-1}$ for 1993-2016, also depending on the dataset choice and location.

For most regions of the world, we find that the GRD-induced SLC trend is higher than the 1-sigma total uncertainty, with exception of the regions near the polar areas (indicated by stipples in Figure 7). Comparing the JPL trend to the IMB+WGP trend, the shape of the pattern is similar, but the global mean (and thereby the regional SLC) is larger for the IMB+WGP combination. Nonetheless, both distributions of the regional SLC have a similar upper bound, with the 90%-range of the ocean grids ranging from $-0.26$ to $2.24$ mm.year$^{-1}$ and from $-0.43$ to $2.20$ mm.year$^{-1}$, for the JPL and IMB+WGP datasets. The regional histograms also show a skewed distribution of the trend, with mainly positive values. When we compare the two periods of IMB+WGP (Figure 7c, e), the regional histogram is slightly narrower for the longer period (i.e., less divergence for

the regional values), with the 90%-range of the ocean grids ranging from $-0.32$ to $1.50$ mm.year$^{-1}$. This is probably because the local effect of internal variability plays a smaller role in the longer period. Nonetheless, the regional pattern is similar for both periods.

The uncertainty patterns (Figure 7, right panels) are similar for the different dataset combinations (JPL vs. IMB+WGP) and periods (2003-2016 vs. 1993-2016). However, the regional histograms are slightly different, with the 90%-range of the regional uncertainties ranging from $0.89$ to $1.32$ mm.year$^{-1}$ and from $0.63$ to $0.98$ mm.year$^{-1}$, for respectively JPL and IMB+WGP for the 2003-2016 period. Similar to the trend, the longer period IMB+WGP uncertainties have a similar pattern but with lower values than for the shorter period, with regional values ranging from $0.38$ to $0.60$ mm.year$^{-1}$. Although the total uncertainty is dominated by the temporal uncertainty (see Figure 8), the similarity of the uncertainty pattern for both periods is influenced by the fact that the spatial-structural errors are based on the 2003-2016 period and extended to 1993-2016. On average, the spatial-structural uncertainty represents $14\%$ ($21\%$) of the total uncertainty, while the temporal represents $77\%$ ($75\%$), for the 2003-2016 (1993-2016) period.

### 3.6 Coastal Examples

To further illustrate how the different contributions and uncertainties contribute to the total uncertainty budget, we selected ten coastal cities around the world in which we break down the total uncertainty of GRD-induced SLC from 1993-2016 into the four contributions (Figure 8a), and into the three types of uncertainties (Figure 8b). We also show the different types of uncertainties for each of the contributions (Figure 8c). As in in Figure 7, we show the IMB+WGP combination.

The large contribution of the LWS and temporal uncertainty to the uncertainty budget is highlighted on Figure 8. Figure 8a shows that the LWS uncertainty plays an important role at all locations, being responsible for at least $50\%$ of the total uncertainty. While the temporal uncertainty is the main contribution of the LWS uncertainty (Figure 8c), in some locations, such as Vancouver (Canada, location 1), Washington (US, location 3) and Tokyo (Japan, location 9) the spatial uncertainty is also important. Even without the contribution of LWS to the total uncertainty (Supplementary Figure A7b), the temporal uncertainty is still the main contributor. The intrinsic uncertainty (panel b) is fairly small in all locations, with an average contribution of $8\%$ for this dataset combination. However, for the JPL combination (Supplementary Figure A6), which has intrinsic uncertainty estimation for all contributions, the intrinsic uncertainty is responsible, on average, for $30\%$ of the total uncertainty, being more important than the spatial-structural one.

The second main contribution to the uncertainty budget comes from the AIS, except for Vancouver (Canada, location 1), for which the glaciers (GLA) contribute about 2 times more than AIS. The AIS uncertainty is mainly dominated by the intrinsic uncertainty, with exception of Cape Town (South Africa, location 6), which is located within the large uncertainty contours of the spatial-structural uncertainty from AIS (see Figure 5a). In general, the relative importance of GIS and GLA is fairly similar, with exception of Vancouver (Canada, location 1) and Rotterdam (the Netherlands, location 5). In such locations, the GLA uncertainty is dominated by the spatial-structural contribution, while in all other locations the temporal uncertainty plays the most important role. On average, the GIS uncertainty is dominated by the intrinsic and temporal uncertainties rather than by spatial-structural uncertainty (panel c).

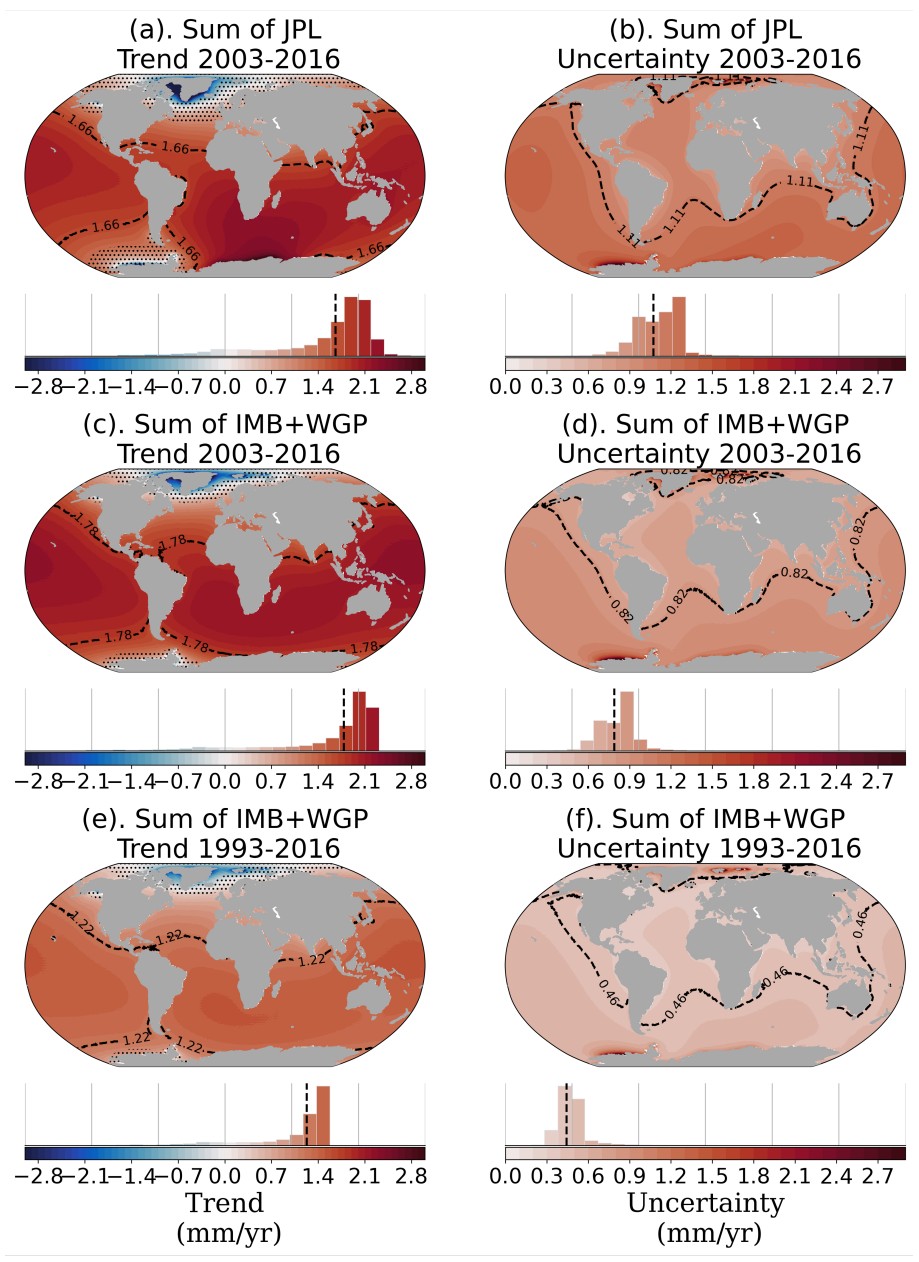

**Figure 7.** Total GRD-induced SLC fields of the trend and uncertainty $(\mathrm{mm.year^{-1}})$ (AIS+GIS+LWS+Glaciers contributions; intrinsic + temporal + spatial uncertainties) for GRACE (a,b) and IMBIE+WaterGAP for 2003-2016 (c,d) and for 1993-2016 (e,f). Histograms underneath each map indicates the distribution of the regional values across the oceans, in which the 5 to 95th percentile range (90%-range) is based on. Spatial average of the regional trend and uncertainty indicated by black dashed lines in the maps and bar charts. Regions with trends smaller than the 1-sigma uncertainty are indicated in the map with stipples.

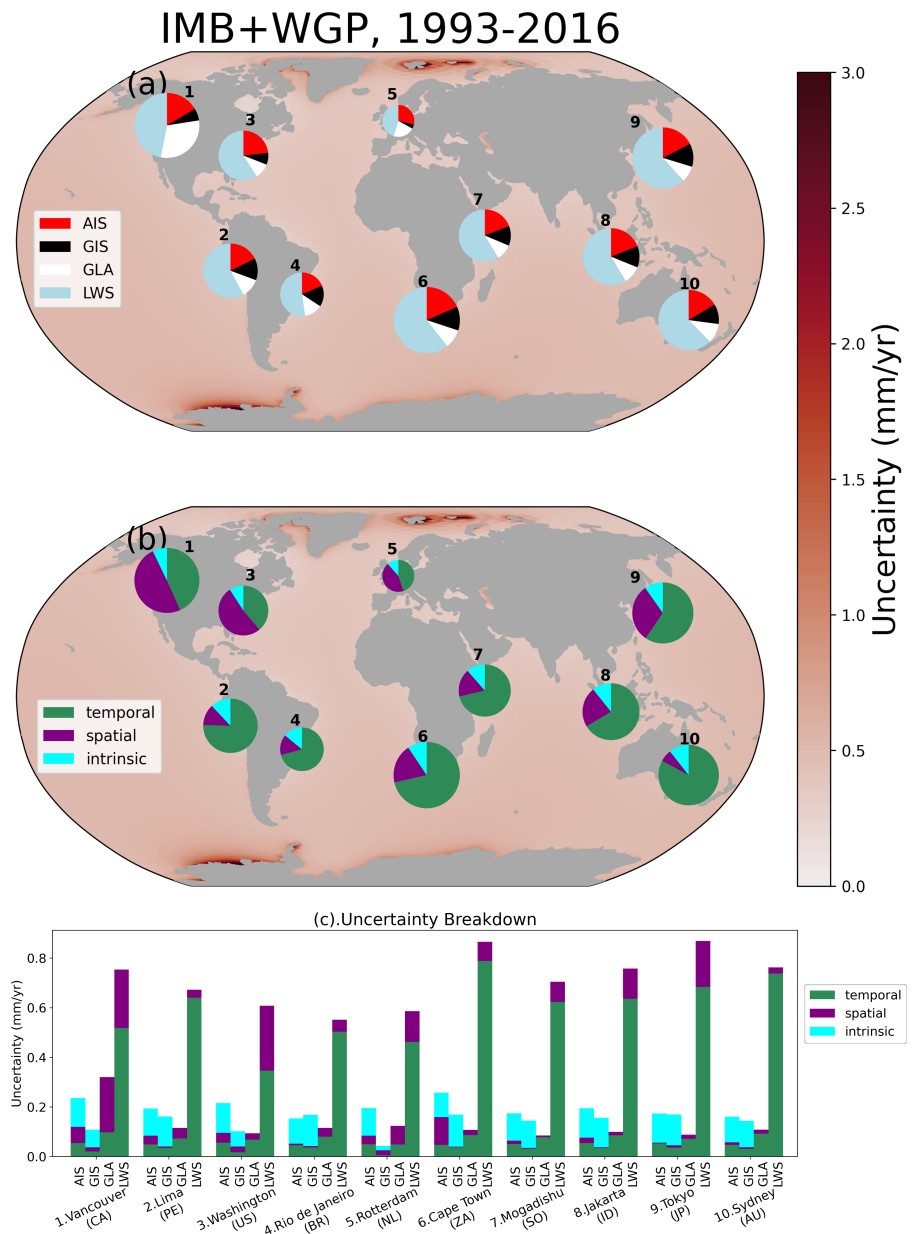

**Figure 8.** Pie charts represent the total uncertainty separated by (a) contribution and (b) type of uncertainty, and the bars the breakdown for each contribution (c). Background maps show the total GRD-induced uncertainty. The size of the pie charts is relative to the magnitude of the total uncertainty. Note that the uncertainties are combined in quadrature, so simply adding up the bars in panel c will not reflect the size of the pie charts on panels a and b.

## 4   Discussion and Conclusion

In this manuscript we investigated the regional GRD-induced SLC patterns associated with barystatic contribution to sea-level trends over 1993-2016 and 2003-2016, focusing on improving the understanding of the uncertainty budget. We showed how mass changes of glaciers, land water storage, and the Greenland and Antarctic ice sheets influence regional SLC by computing sea-level fingerprints. We considered three types of uncertainties in our budget: the determination of a linear trend (temporal); the spread around a central estimate as influenced by the distribution of mass change sources (spatial); and the uncertainty from the data/model itself (intrinsic).

The uncertainty budget is dominated by the temporal uncertainty,responsible on average for $65\%$ of the total uncertainty, while the spatial-structural and intrinsic uncertainties have smaller contributions of similar magnitude, responsible on average for $16\%$ and $18\%$ of the budget, respectively. The temporal uncertainties associated with the trend may represent real climatic signals, and not only measurement errors. For example, the variability due to climatic oscillations, such as El Niño Southern Oscillation (ENSO) and the Pacific Decadal Oscillation (PDO), may be reflected in the residuals of the time series, affecting the trend and its temporal uncertainties (Royston et al., 2018). As such climatic events influence not only mass change, but also other drivers of sea-level change (e.g., thermal expansion), caution must be taken when using and comparing these uncertainties with those from other sea level contributors. Despite the dataset-driven differences, for a given contribution all estimated trends agree within their respective 1-sigma uncertainties, both for regional and global mean values (Figure 1), Supplementary table A1).

We find that the total GRD-induced sea-level trends range from $-0.43$ to $2.20$ mm.year$^{-1}$ for 2003-2016, and from $-0.32$ to $1.50$ mm.year$^{-1}$ for 1993-2016, depending on location, for the IMB+WGP combination, with spatial averages of $1.78$ and $1.22$ mm.year$^{-1}$, respectively. The total uncertainty of the GRD-induced sea-level trend ranges from $0.63$ to $0.98$ mm.year$^{-1}$ for 2003-2016, and from $0.38$ to $0.60$ mm.year$^{-1}$ for 1993-2016 for the IMB+WGP combination, with spatial averages of $0.80$ and $0.46$ mm.year$^{-1}$, respectively. While these uncertainty values may seem large compared to studies focusing on global changes alone (Horwath et al., 2021; Frederikse et al., 2020), other studies also found that regional uncertainties are higher than the previously published global mean rates (Prandi et al., 2021; Bos et al., 2014). For example, in a recent satellite altimetry sea-level change assessment, Prandi et al. (2021) found that the local sea-level trend uncertainty due to observational errors (i.e., intrinsic uncertainties) was about two times higher than the global mean sea-level trend uncertainty of Ablain et al. (2019). We note that the spatial average of the regional uncertainties (indicated by the black dashed line in the figures) is not equal to the uncertainty of the global mean barystatic SLC time series and trend. Consequently, the spatial averages will lead to larger values then the uncertainty of the global mean sea-level time series (see Figure A5). Thus, one should not compare the value given here to characterize global mean sea-level changes with other studies focusing on the global mean (e.g. Horwath et al. (2021)).

The GRD-induced sea-level trends clearly show the classical gravitational-rotational-deformational pattern, matching qualitatively with other fingerprints (e.g., Mitrovica et al. (2001); Riva et al. (2010); Hsu and Velicogna (2017); Jeon et al. (2021)). Our spatial-structural uncertainties highlight the effect of using a uniform mass change (i.e., only one value averaged over a

region) compared to non-uniform local mass changes (Bamber and Riva, 2010; Mitrovica et al., 2011). For example, we show that different location of mass changes can lead to deviations larger than 20% for AIS (Figure 5). As a consequence of the relatively low spatial resolution of the observations, the AIS is the second main contributor to the total GRD-induced uncertainty budget. We show that this effect is important not only for AIS, but for all the GRD-induced SLC contributions.

The main source of uncertainty in the GRD-induced SLC is the temporal uncertainty from the land water storage (LWS) contribution, which is responsible for $35 - 60\%$ of the total uncertainty, depending on the region of interest. . This is likely related to the (climate-driven) natural variability of LWS (Vishwakarma et al., 2021; Hamlington et al., 2017; Nerem et al., 2018), which is mainly driven by seasonal and interannual cycles (Cáceres et al., 2020). A method to deal with the natural variability of LWS would be to use different metrics than linear trends (Vishwakarma et al., 2021), such as time varying trends based on a state space model (Frederikse et al., 2016; Vishwakarma et al., 2021). However, we choose to use linear trends in this study for the sake of accuracy, reproducibility and discussion. It has also been suggested that a more appropriate way of computing a meaningful linear trend from LWS is to incorporate this variability in the analysis (Vishwakarma et al., 2021), as we did by including the seasonal components in the functional model. Nonetheless, the LWS uncertainties related to the trend are still very high, suggesting that a period of 25 years (1993-2016) might still be too short to solve the low frequency natural variability of LWS, particularly on (multi)-decadal timescales. Indeed, Humphrey et al. (2017) showed that removing the short-term climate-driven variability of the LWS signal yields in a more robust long-term (>10 years) trend, with reduced uncertainties.

In this study we assessed the uncertainties related to the regional GRD-induced patterns associated with barystatic sea-level change, in particular their spatial distribution. The true uncertainty of ocean mass contribution to sea-level change is difficult to determine. Our approach of quantifying this uncertainty is to some extent conservative, as it results in larger uncertainties than in previous studies (e.g., Horwath et al. (2021)). Nonetheless, we did assume independence of the different types of uncertainty, and did not propagate GIA uncertainties into our fingerprints, which could lead to even larger uncertainties. Our results highlight that improving the spatial detail of land ice mass loss products, as well as determining more accurate land water storage trends, would lead to better SLC estimates. In addition, our findings can be used to inform projection frameworks. For example, we show that the distribution of ice in the Antarctic Ice Sheet has a significant impact on regional SLC, even in locations far from the ice sheets, such as the Netherlands. This means that, depending on the region of a collapse in the Antarctic Ice Sheet, the sea-level rise projections, which are often based on uniform ice sheet distributions and static fingerprints (e.g., Slangen et al. (2012); Jevrejeva et al. (2019)) , may have large regional deviations due to spatial differences in the mass source. Incorporating the insights of uncertainty assessments in sea-level frameworks (as in Larour et al. (2020)) should eventually lead to better sea-level projections.

*Code and data availability.* The data used in this manuscript is available at 4TU database (https://doi.org/10.4121/16778794). The code for generating the figures is available at github repository https://github.com/carocamargo/barystaticSLC.

**Appendix A:  Data Description**

The datasets used in this manuscript are briefly described below. In-depth description of each dataset can be found in their respective references.

**A1   GRACE Mascon Estimates**

We use GRACE land mass concentrations (mascons) solutions from two processing centres: RL06 v02 from CSR (Save et al., 2016; Save, 2020) and RL06 v02 from JPL (Watkins et al., 2015; Wiese et al., 2019). We chose to use the mascons solution
instead of spherical harmonics to avoid the land-ocean leakage issue (Jeon et al., 2021; Chambers et al., 2007). The mascons include all mass changes in the Earth system, accounting for variations in land hydrology and in the cryosphere, as well as solid Earth motions (Adhikari et al., 2019). We do not, however, use the changes in the ocean, since we focus on land hydrology and cryosphere variations. CSR and JPL mascons are provided on a 0.25 and 0.5 degree grids, respectively, even though the native resolution of the GRACE/GRACE-FO data is roughly 300km (i.e., 3-degree equal-area mascons). The native resolution of
CSR mascons are $1°x1°$ equal-area grid and and $3°x3°$ for JPL mascons. Since the native resolution of GRACE observations of about 300 km at the equator (Tapley et al., 2004), the JPL mascons have independent solutions at each mascon centres, with uncorrelated errors, while the CSR mascons are not fully independent and are expected to contain spatially correlated errors. Both mascons have been corrected for glacial isostatic adjustment (GIA) with the ICE6G-D model (Peltier et al., 2018), and for ocean and atmosphere dealiasing (AOD1B 'GAD' fields). In addition, the JPL mascons use a Coastline Resolution
Improvement (CRI) filter to separate land/ocean mass within the mascon (Wiese et al., 2016). Only the JPL mascons are provided with intrinsic uncertainty estimates (Wahr et al., 2006; Wiese et al., 2016). Both mascons are given with a monthly frequency, ranging from April-2002 to August-2020.

**A2   IMBIE Estimates**

For both ice sheets we use the products of IMBIE (Shepherd et al., 2018, 2020), which combines several estimates (26 for GIS
and 24 for AIS) of ice sheet mass balance derived from satellite altimetry, satellite gravimetry and the input-output method. The monthly datasets cover the period 1992-2017 and 1993-2018 for AIS and GIS, respectively. In addition to the total ice sheet mass balance, the GIS dataset also distinguishes between surface mass balance (GRE SMB) and dynamic ice discharge (GRE DYN). For the AIS, the data is subdivided in the main 3 drainage regions: West Antarctica, East Antarctica and the Antarctic Peninsula. The IMBIE estimates are provided with intrinsic uncertainty estimates, reflecting the combination of
several different datasets.

**A3   UCI AIS and GIS Estimates**

Using improved records of ice thickness, surface elevation, ice velocity and a surface mass balance model (RACMOv2.3), Mouginot et al. (2019) and Rignot et al. (2019) present yearly reconstructions of mass changes from the 1970s until 2017 and 2018 for the Greenland and Antarctic ice sheets, respectively. These GRACE-independent reconstructions agree, within

475 uncertainties, with estimates from radar and laser altimetry and GRACE. The reconstructions are provided as the mean for each drainage basin, based on ice velocity data (18 basins for AIS (Rignot et al., 2011) and 6 for GIS (Mouginot and Rignot, 2019)).

## A4 WaterGAP Hydrological Model

We use the integrated version of the WaterGAP global hydrological model (Döll et al., 2003) v2.2d with a global glacier model
(Marzeion et al., 2012), presented in Cáceres et al. (2020). The hydrological model uses a homogeneized climate forcing from WFDEI (Weedon et al., 2014), with the precipitation correction of GPCC (Schneider et al., 2015). The model is provided on a 0.5 degree grid, covering all continental areas except for Antarctica. In order to consistently treat both ice sheets (GIS and AIS), we remove Greenland from the model. The WaterGAP model simulates human water use, daily water flows and water storage, taking into account dams and reservoirs based on the GRanD database (Lehner et al., 2011) and assuming that consumptive
irrigation water use is 70% of the optimal level in groundwater depletion areas. The glacier model computes mass changes for individual glaciers around the world (based on the Randolph Glacier Inventory (Pfeffer et al., 2014), including glaciers surface mass balance, glacier geometry, air temperature and several others glacier-specific parameters and variables (Marzeion et al., 2012). The dataset is provided at a monthly frequency, from 1948-2016.

## A5 PCR-GLOBWB Hydrological Model

The second global hydrological model included in our analysis is the PCRaster Global Water Balance 2 model (PCR- GLOBW, Sutanudjaja et al. (2018)), which fully integrates different water uses, such as water demand, groundwater and surface water withdrawal, water consumption, with the simulated hydrology. The model is forced with the W5E5 version 1 (Lange, 2019), covering the period 1979-2016. It provides monthly averages of total water storage thickness with a 5 arcmin resolution. Dams and reservoirs form the GRanD database (Lehner et al., 2011) are also included in the model. As this model does not explicitly
resolve glaciers nor includes ice sheets, we mask out all the glaciated areas.

## A6 Zemp 2019 Glacier data

We use the yearly glacier mass loss estimates from Zemp et al. (2019) over the period 1961 to 2016. This dataset combines the temporal variability from the glaciological data, computed using a spatio-temporal variance decomposition, with the glacier-specific values of the geodetic observations. Both glaciolocial and geodetic observations comes from the World Glacier
Monitoring Service (WGMS, 2021). This combined data is then statistically extrapolated to the full glacier sample to assess regional mass changes, taking into account regional rates of area change. This dataset provides regional mass changes for the 19 regions of the Randolph Glacier Inventory (Consortium, 2017; Pfeffer et al., 2014). As the IMBIE estimates already account for peripheral glaciers to the ice sheets, we remove these from the Zemp dataset.

**Appendix B: Supplementary Figures and Tables**

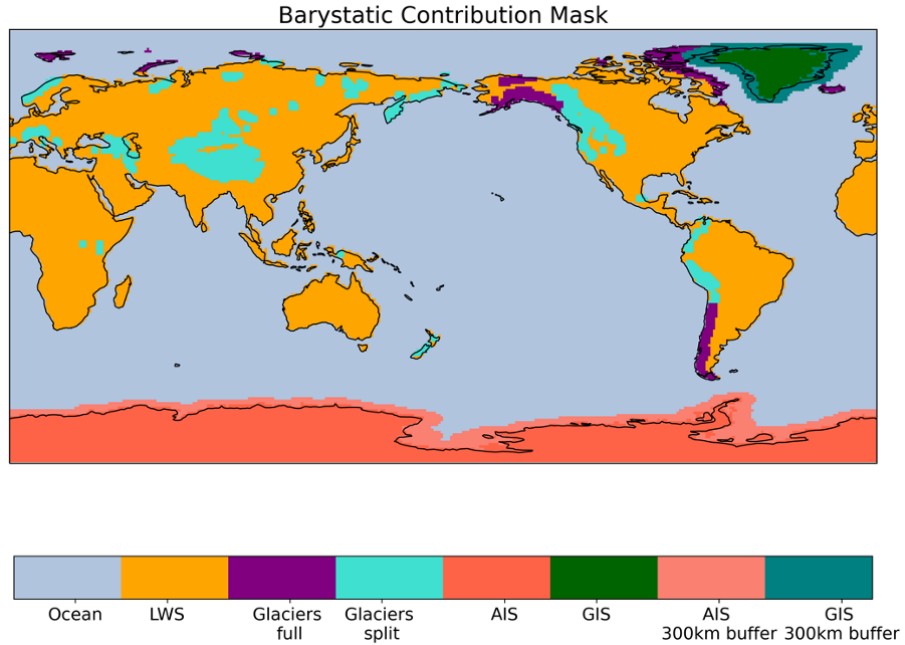

**Figure A1.** Mask of the different contributions to barystatic sea-level change.

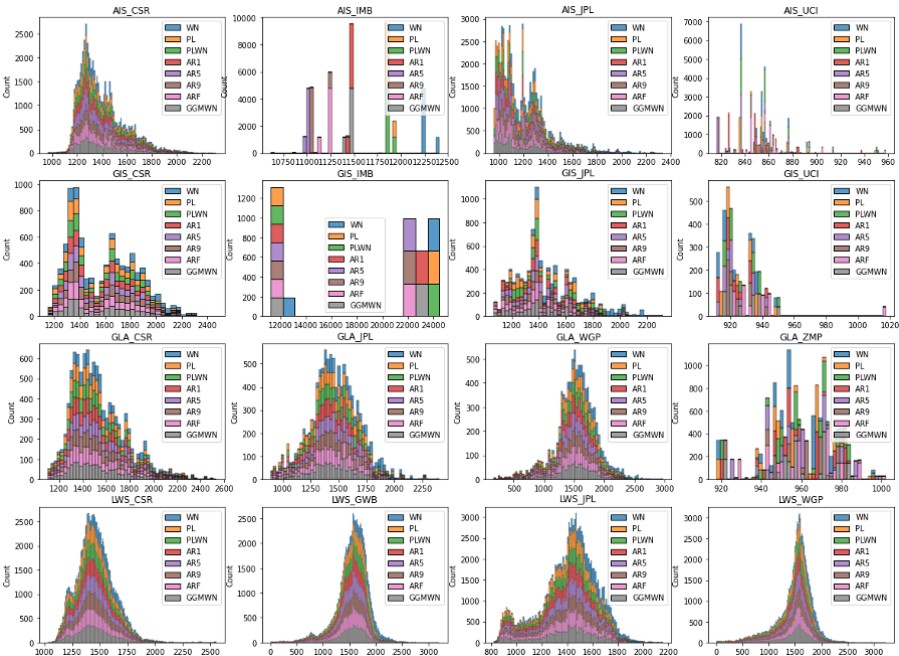

**Figure A2.** Histogram of the modified Bayesian Information Criterion for each dataset, used to select the optimal noise models. The x-axis shows the BIC score, and the y-axis the number of grid points (count). Note that all models have scores within the same range, showing that no model fails in capturing the signal of the observation.

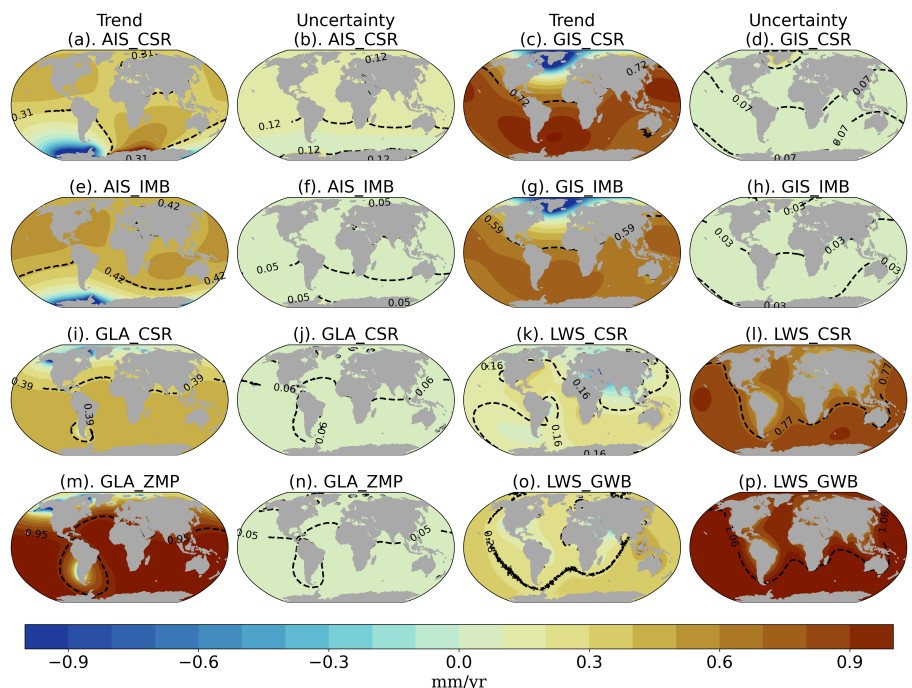

**Figure A3.** GRD-induced sea-level trend and temporal uncertainty ($\mathrm{mm.year}^{-1}$) for GRACE (CSR) and independent combination (IMB + ZMP + GWB) for 2003-2016. Black dashed contour line and number indicates the spatial average of the regional trend and uncertainty. Complementary of trends and uncertainties of Figure 4.

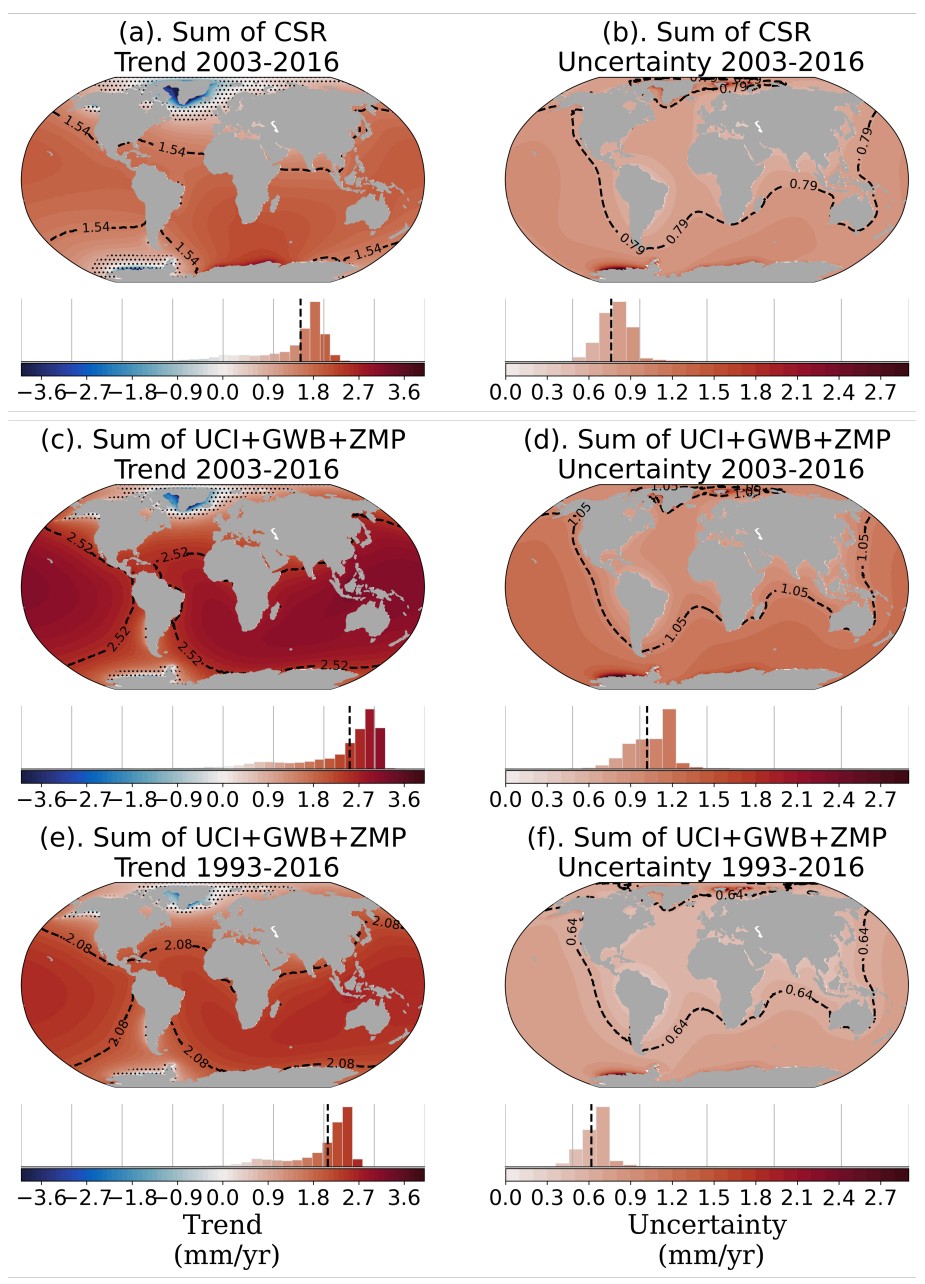

**Figure A4.** Total GRD-induced SLC fields of the trend and uncertainty ($\text{mm.year}^{-1}$) (AIS+GIS+LWS+Glaciers contributions; intrinsic + temporal + spatial uncertainties) for GRACE CRS (a,b) and UCI + GlobWEB + Zemp for 2005-2015 (c,d) and for 1993-2016 (e,f). Histograms underneath each map indicates the distribution of the regional values across the oceans. Spatial average of the regional trend and uncertainty indicated by black dashed lines in the maps and bar charts. Complementary of trends and uncertainties of Figure 7.

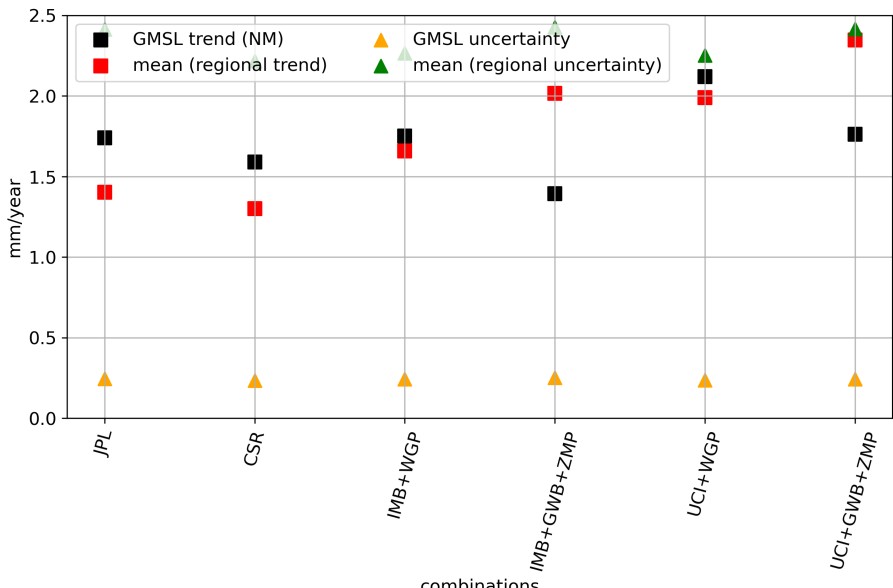

**Figure A5.** Comparison of global mean sea-level trend (black squares) and uncertainty (yellow traingles) with the spatial average of the regional trend (red circles) and uncertainty (green upside down triangles) from 2003-2016. The difference between the GMSL trend and spatial average of the regional trend is due to the use of regionally different noise models (following selection of Figure 3)

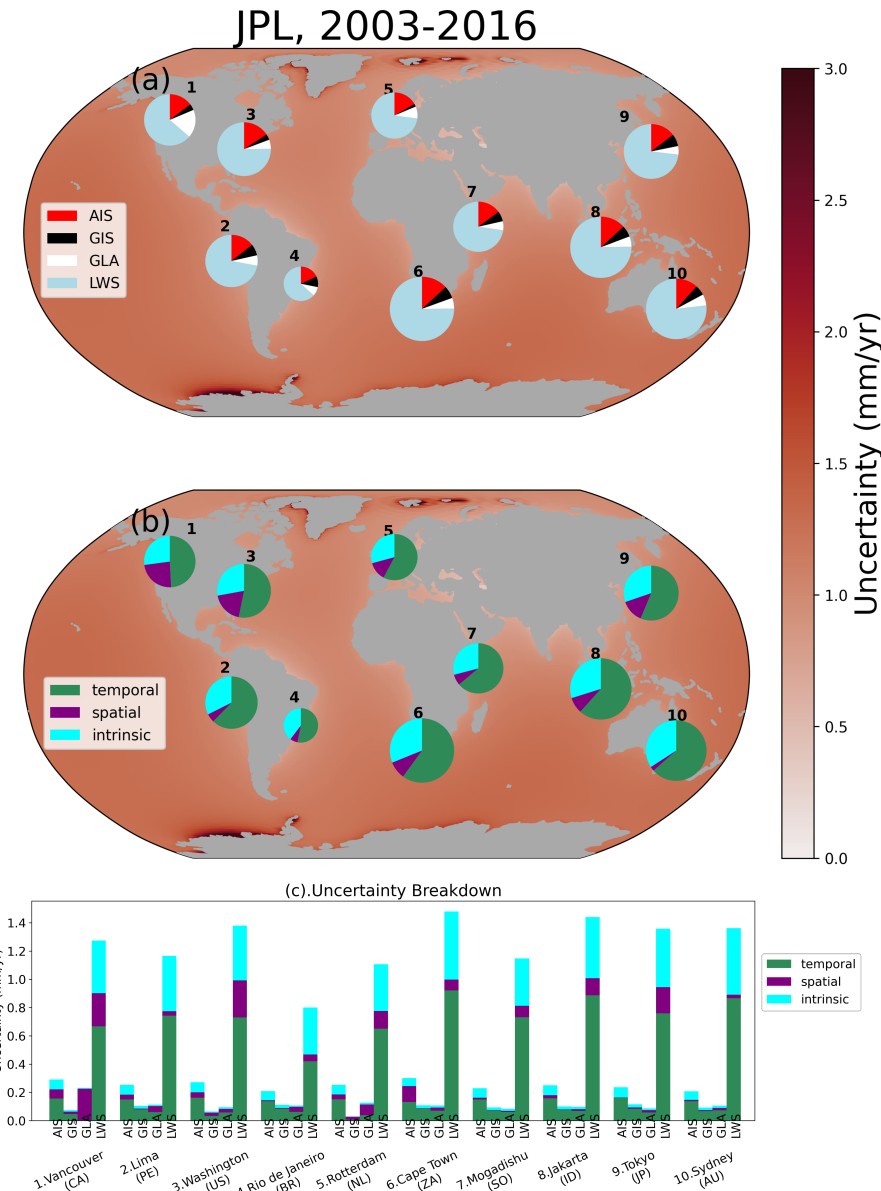

**Figure A6.** Same as Figure 8, for JPL dataset, from 2003-2016.

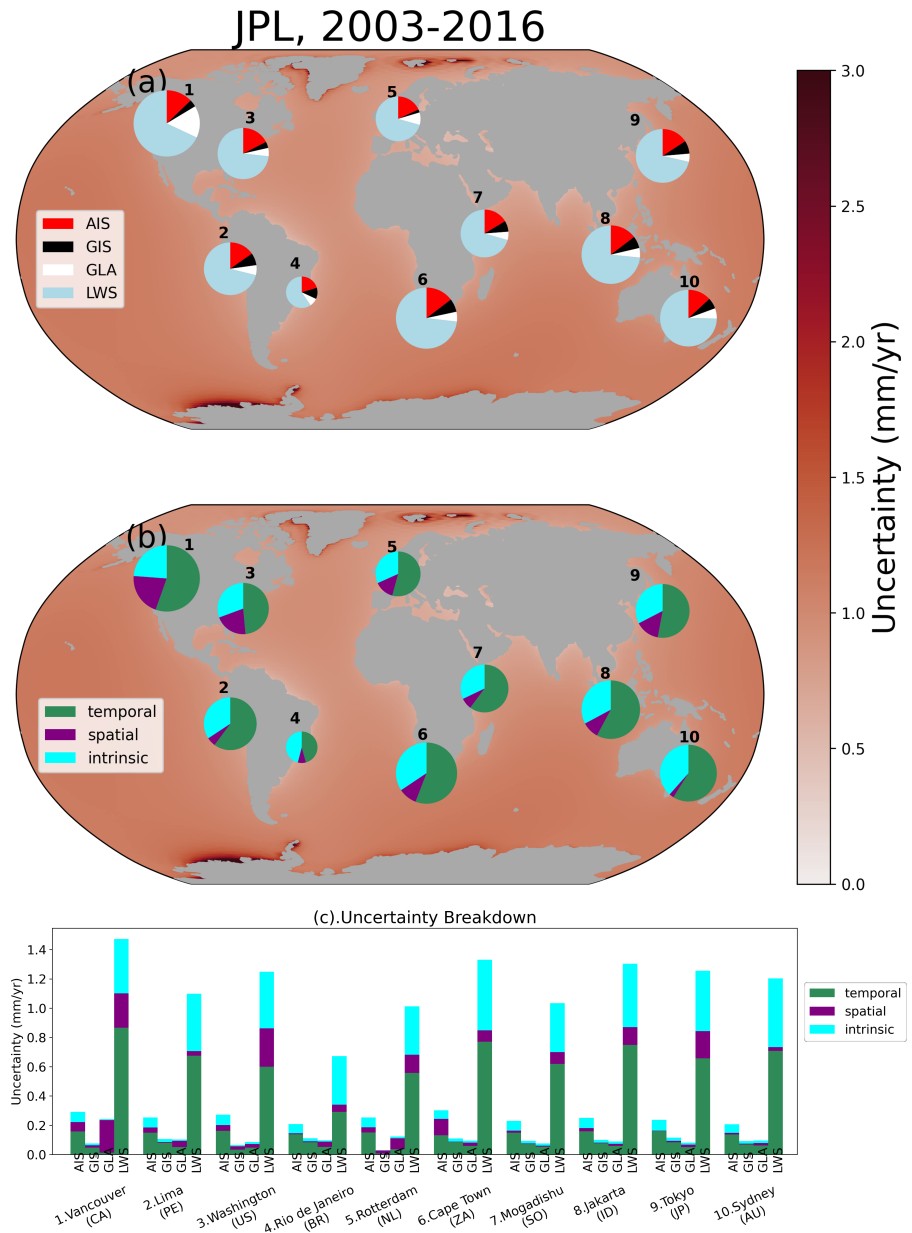

**Figure A7.** Same as Figure 8, but without the contribution of land water storage (LWS)

**Table A1.** Global mean barystatic sea-level contributions and uncertainties. Note that these numbers may be different compared to the histograms of Figure 7, which represent the spatial average of the regional trend and uncertainty. The difference between the trends is due to the use of noise-models for the regional trend, against an ordinary least-squares fit for the global mean trend. Note that we remove the 'spatial' part of the spatial-structural uncertainty of the regional assessment, and define the structural uncertainty as the standard deviation of the trends for the same contribution.

| | 2003-2016 | | | | | | 1993-2016 | | | | | |
|---|---|---|---|---|---|---|---|---|---|---|---|---|
| | trend | ± | $\sigma_{total}$ | $\sigma_{temporal}$ | $\sigma_{structural}$ | $\sigma_{intrinsic}$ | trend | ± | $\sigma_{total}$ | $\sigma_{temporal}$ | $\sigma_{structural}$ | $\sigma_{intrinsic}$ |
| **AIS** | | | | | | | | | | | | |
| AIS_CSR | 0.32 | ± | 0.09 | 0.03 | 0.09 | | | | | | | |
| AIS_JPL | 0.27 | ± | 0.1 | 0.04 | 0.09 | 0.04 | | | | | | |
| AIS_IMB | 0.37 | ± | 0.13 | 0.05 | 0.09 | 0.07 | 0.19 | ± | 0.15 | 0.04 | 0.14 | 0.03 |
| AIS_UCI | 0.48 | ± | 0.09 | 0.01 | 0.09 | | 0.4 | ± | 0.14 | 0.01 | 0.14 | 0.03 |
| **GIS** | | | | | | | | | | | | |
| GIS_CSR | 0.72 | ± | 0.32 | 0.03 | 0.31 | | | | | | | |
| GIS_JPL | 0.73 | ± | 0.32 | 0.03 | 0.31 | 0.01 | | | | | | |
| GIS_IMB | 0.53 | ± | 0.32 | 0.03 | 0.31 | 0.07 | 0.36 | ± | 0.12 | 0.03 | 0.11 | 0.03 |
| GIS_UCI | 0.06 | ± | 0.32 | 0.08 | 0.31 | | 0.52 | ± | 0.12 | 0.03 | 0.11 | |
| **GLA** | | | | | | | | | | | | |
| GLA_CSR | 0.68 | ± | 0.16 | 0.06 | 0.15 | | | | | | | |
| GLA_JPL | 0.64 | ± | 0.16 | 0.07 | 0.15 | 0.01 | | | | | | |
| GLA_WGP | 0.58 | ± | 0.15 | 0.03 | 0.15 | | 0.51 | ± | 0.16 | 0.03 | 0.16 | |
| GLA_ZMP | 0.92 | ± | 0.15 | 0.03 | 0.15 | | 0.74 | ± | 0.17 | 0.04 | 0.16 | |
| **LWS** | | | | | | | | | | | | |
| LWS_CSR | 0.09 | ± | 0.14 | 0.12 | 0.06 | | | | | | | |
| LWS_JPL | 0.22 | ± | 0.33 | 0.12 | 0.06 | 0.3 | | | | | | |
| LWS_WGP | 0.20 | ± | 0.12 | 0.1 | 0.06 | | | ± | 0.07 | 0.04 | 0.06 | |
| LWS_GWB | 0.18 | ± | 0.12 | 0.1 | 0.06 | | | ± | 0.07 | 0.04 | 0.06 | |
| **Combination** | | | | | | | | | | | | |
| CSR | 1.81 | ± | 0.39 | 0.14 | 0.36 | | | | | | | |
| JPL | 1.86 | ± | 0.49 | 0.15 | 0.36 | 0.3 | | | | | | |
| IMB+WGP | 1.68 | ± | 0.39 | 0.12 | 0.36 | 0.1 | 1.27 | ± | 0.26 | 0.07 | 0.25 | 0.04 |
| IMB+GWB+ZMP | 2.00 | ± | 0.39 | 0.12 | 0.36 | 0.1 | 1.58 | ± | 0.26 | 0.08 | 0.25 | 0.04 |
| UCI+WGP | 1.32 | ± | 0.38 | 0.13 | 0.36 | | 1.64 | ± | 0.25 | 0.06 | 0.25 | |
| UCI+GWB+ZMP | 1.64 | ± | 0.38 | 0.13 | 0.36 | | 1.95 | ± | 0.26 | 0.06 | 0.25 | |

*Author contributions.* CC performed the research and drafted the article. CC, RR and AS designed the study. All authors contributed to the interpretation of the results and the writing of the manuscript.

*Competing interests.* The authors declare that they have no conflict of interest.

*Acknowledgements.* We thank Thomas Frederikse and the anonymous reviewer for their helpful comments. This research was funded by the Netherlands Space Office User Support program (grant no. ALWGO.2017.002). All figures were done in python, using scientific colour
maps from Crameri (2018) and from Thy (2016).

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
