# Peer review of "Trends and Uncertainties of Mass-driven Sea-level Change in the Satellite Altimetry Era"

_Earth System Dynamics, 2021_

## Referee Comment (RC2)

Review of 'Trends and Uncertainties of Regional Barystatic Sea-level Change in the Satellite Altimetry Era'

In this manuscript, the authors have collected a large dataset of the sources of barystatic sea-level changes, computed the associated GRD fingerprints, and discuss the sources of uncertainty in the resulting GRD pattern. I think this dataset will be very useful for many applications in sea-level and climate research, and I'm convinced this work is a worthy contribution to the literature on this topic.

Nevertheless, I have a few technical and philosophical remarks and questions. I start with some general remarks, followed by line-by-line comments.

Thomas Frederikse

**General remarks**

GR1 Intrinsic uncertainties:
The reported uncertainties from the individual data sets are assumed to be uncorrelated. Therefore, the trends are subsequently estimated using ordinary least squares with the uncertainties on the diagonal of the co-variance matrix. I don't think that this is the right approach to estimate the trend uncertainties due to intrinsic errors. For many estimates the uncertainties will be serially correlated, for example due to uncertainties that affect the trend. Like if the uncertainty in a GRACE time series is fully due to GIA uncertainties that only affect the trend, the aforementioned assumption doesn't hold, and the resulting errors are an underestimation. For estimates where the rates and their uncertainties are available, an approximation that often works well is to assume that the rate errors are uncorrelated, which is equal to assuming random walk. Then you can generate an ensemble of time series by perturbing the rate with random normal noise, and then integrate the rates to obtain the time series. Something like:

```
rate[t] # rate of a process, say in GT/yr
rate_unc[t] # uncertainty of the rate
for ens in 1:1000 # Let's make 1000 ensembles
  rate_ens = rate + rand_normal[length(t)]*rate_unc
  tseries_ens = cumsum(rate_ens) # This gives the time series in GT
end
```

can do the trick. A good way to verify the results is to compare the barystatic trend uncertainties in Gt/yr or mm/yr with those reported in the papers where the data sets came from.

GR2 Trend uncertainties:
I think the paper could use some more discussion on the meaning of the trend uncertainties, because their meaning is not trivial and explaining the meaning of these trend uncertainties is important for data users to correctly apply them. This is a bit of a philosophical point, but the auto-correlated residuals after estimating the trend are not per se due to measurement errors, but they could represent a real signal. An example is the drop in GMSL during the 2010/2011 La Nina event, and the acceleration in ice-sheet mass loss. Let's assume now that someone downloads the regional data as well as some altimetry data of regional sea level. Then the uncertainty in the unexplained residual (Local altimetry – local ocean mass) should not contain the trend

uncertainty given here. Altimetry will also see the acceleration and La Nina-like bumps and throughs and the difference probably shows much less interannual variations and thus has a lower trend uncertainty. However, when this user just uses the provided trend uncertainty and adds it in quadrature to some other errors, he or she will overestimate the uncertainties of the just computed difference. Some guidance could help here.

GR3 Barystatic sea level:
How do the global-mean time series and uncertainties look like? A simple plot with the global-mean time series from each component and the total might be a nice addition. That could also help verifying the 1993-2016 time series from models: do they show similar trends and variability as GRACE? For TWS, Scanlon et al. (2018) show some discrepancies between models and GRACE for TWS. Might be interesting to see whether estimates from WaterGAP and PCR-GLOBWB now perform better.

GR4 The term 'regional barystatic':
After Gregory et al. (2019), barystatic sea level should only refer to global-mean sea-level changes, and not to regional patterns. I'd remove the term 'regional barystatic' and replace it with 'Barystatic sea level and associated GRD patterns' or something like that

GR5 Glaciers and ice sheets:
A nasty problem when working with GRACE data for glaciers is that the GRACE resolution is pretty coarse compared to the size of some glaciers. Therefore, it's hard to separate the mass loss from peripheral glaciers in Greenland and Antarctica from the nearby ice sheets. The same goes for small glaciers in for example Asia, where mass changes from nearby TWS changes leak into the glacier mass change estimates. Did the authors take this into account? A possible way out is the method described in the supporting information of Reager et al. (2016): for the RGI regions where glacier mass loss dominates the GRACE mass change estimates, use GRACE, for the ice sheets, treat the glaciers as part of the ice sheets, and for the other regions, use another dataset (for example Malles & Marzeion, 2021 or Hugonnet et al. 2021) to separate glaciers from TWS. We have used such an approach in a recent paper on ocean heat content (Hakuba et al. 2021), and the scripts to do such a separation can be found on Github: look at the function `separate_mass_ctb` in https://github.com/thomasfrederikse/EEI_GRACE/blob/main/Code/mod/mod_budget_grace_mass_rsl_ens.py.

GR6 Used datasets.
The authors have collected a large and diverse set of sources for their barystatic estimates. I've listed below a few other data sets that could be added as well. Since new data sets appear all the time, this list isn't exhaustive and should be seen more as an idea rather than a demand to incorporate them in this manuscript.

Glaciers:
- Model estimates from Malles & Marzeion (2021). This data set also contains some estimates of the intrinsic errors due to model choices and input dataset.
- Satellite altimetry observations from Hugonnet et al. (2021).

Greenland:
- The model estimates from Mankoff et al. (2021)

Antarctica, Greenland and Glaciers:
- Bamber et al. (2018) provides an IMBIE-like assessment of mass loss from glaciers and ice sheets.

Terrestrial Water Storage
- Humphrey and Gudmundsson (2019) have TWS estimates based on a simple bucket approach trained with GRACE data. It comes with an ensemble from which uncertainties can be deducted.

**Line-by-line comments**

L10: the trend ranges, do they refer to the 95$^{th}$ percentile of the gridded field? I'd think the minimum trend will be lower very close to the ice sheet edges

L63-L64: "The structural uncertainty is related to the use of different datasets of the same process". The structural and intrinsic uncertainties, are they independent? I can imagine that for example in GRACE, there's an uncertainty in some atmospheric correction, and product A uses estimate A and product B uses estimate B for that process. Then parts of the intrinsic uncertainties also end up in the structural uncertainties.

L131: More of an idea than a comment: there's a lot of people looking for fingerprints to analyze altimetry data, so there might be quite some interest in complementary geocentric sea-level fingerprints.

L166 Average – typo

L179 The JPL mascon uncertainties do not represent the uncertainties due to the GIA correction. This uncertainty is actually pretty large, but a bit cumbersome to propagate into the final GRD patterns. It can be done by using uncertainty estimates from for example Caron et al. (2018) and Simon & Riva (2020, I'm sure some of the authors are aware of this study). Propagating the full GIA uncertainties into the fingerprints might be a bit too far-fetched for the current manuscript, but it's a good idea to mention that there's additional uncertainty related to GIA in these GRACE estimates.

Figure 2 It looks like the glacier mass balance from the CSR and JPL mascons has been estimated by splitting up some of the mascons. This is a bit tricky: for some regions, the mass changes of the whole mascon are dominated by small glaciers, and taking a part of the mascon induces an error. The opposite also happens. I'd recommend to not split mascons into smaller pieces. See also GR5 for a possible way out.

L255: Just out of curiosity: does the UCI dataset show any mass gains in East Antarctica?

L264: as a rule of thumb, the individual mascons from the JPL solution are all independent and agree more-or-less with the spatial resolution of the GRACE measurements. For other mascons, like GSFC and CSR ones that have a much higher resolution, the individual mascons are not fully independent of eachother.

Figure 4: This is a very interesting figure! I discovered a lot of intriguing phenomena when looking at it.

Figure 5: also related to GR1. If you check the uncertainties listed in Table 1 from the IMBIE Antarctic paper, the reported uncertainties, which are about 50 Gt yr-1 for 5-year periods, seem to be much higher than reported here. This is probably related to the assumption of uncorrelated uncertainties. Using the rates+uncertainties procedure from GR1 might solve this difference.

L375: Check the paper from Humphrey and Gudmundsson (2019), who provide some centennial estimates of TWS changes.

L381: Antarctica Ice Sheet typo

L429 Individuals typo

**References**

Bamber, J. L., Westaway, R. M., Marzeion, B., & Wouters, B. (2018). The land ice contribution to sea level during the satellite era. Environmental Research Letters, 13(6), 063008. https://doi.org/10.1088/1748-9326/aac2f0

Caron, L., Ivins, E. R., Larour, E., Adhikari, S., Nilsson, J., & Blewitt, G. (2018). GIA Model Statistics for GRACE Hydrology, Cryosphere, and Ocean Science. Geophysical Research Letters, 45(5), 2203–2212. https://doi.org/10.1002/2017GL076644

Gregory, J. M., Griffies, S. M., Hughes, C. W., Lowe, J. A., Church, J. A., Fukimori, I., Gomez, N., Kopp, R. E., Landerer, F., Cozannet, G. L., Ponte, R. M., Stammer, D., Tamisiea, M. E., & van de Wal, R. S. W. (2019). Concepts and Terminology for Sea Level: Mean, Variability and Change, Both Local and Global. Surveys in Geophysics. https://doi.org/10.1007/s10712-019-09525-z

Hakuba, M. Z., Frederikse, T., & Landerer, F. (2021). Earth's Energy Imbalance from the ocean perspective (2005 - 2019). Geophysical Research Letters. https://doi.org/10.1029/2021GL093624

Hugonnet, R., McNabb, R., Berthier, E., Menounos, B., Nuth, C., Girod, L., Farinotti, D., Huss, M., Dussaillant, I., Brun, F., & Kääb, A. (2021). Accelerated global glacier mass loss in the early twenty-first century. Nature, 592(7856), 726–731. https://doi.org/10.1038/s41586-021-03436-z

Humphrey, V., & Gudmundsson, L. (2019). GRACE-REC: A reconstruction of climate-driven water storage changes over the last century. Earth System Science Data, 11(3), 1153–1170. https://doi.org/10.5194/essd-11-1153-2019

Malles, J.-H., & Marzeion, B. (2021). Twentieth century global glacier mass change: An ensemble-based model reconstruction. The Cryosphere, 15(7), 3135–3157. https://doi.org/10.5194/tc-15-3135-2021

Mankoff, K. D., Fettweis, X., Langen, P. L., Stendel, M., Kjeldsen, K. K., Karlsson, N. B., Noël, B., van den Broeke, M. R., Solgaard, A., Colgan, W., Box, J. E., Simonsen, S. B., King, M. D., Ahlstrøm, A. P., Andersen, S. B., & Fausto, R. S. (2021). Greenland ice sheet mass balance from 1840 through next week. Earth System Science Data, 13(10), 5001–5025. https://doi.org/10.5194/essd-13-5001-2021

Reager, J. T., Gardner, A. S., Famiglietti, J. S., Wiese, D. N., Eicker, A., & Lo, M.-H. (2016). A decade of sea level rise slowed by climate-driven hydrology. Science, 351(6274), 699–703. https://doi.org/10.1126/science.aad8386

Scanlon, B. R., Zhang, Z., Save, H., Sun, A. Y., Müller Schmied, H., van Beek, L. P. H., Wiese, D. N., Wada, Y., Long, D., Reedy, R. C., Longuevergne, L., Döll, P., & Bierkens, M. F. P. (2018). Global models underestimate large decadal declining and rising water storage trends relative to GRACE satellite data. Proceedings of the National Academy of Sciences, 115(6), E1080–E1089. https://doi.org/10.1073/pnas.1704665115

Simon, K. M., & Riva, R. E. M. (2020). Uncertainty Estimation in Regional Models of Long-Term GIA Uplift and Sea-level Change: An Overview. Journal of Geophysical Research: Solid Earth. https://doi.org/10.1029/2019JB018983

---

## Author Comment (AC1)

We thank the reviewer for the positive evaluation and valuable comments. We provide a point-by-point response to each comment, with reviewer's comments in black, and author's responses in purple. The modified manuscript text is shown in quotation marks and italic, with additions in bold.

This work compares different estimates of ocean mass contributions to sea level rise. These contributions are themselves derived from different estimates of contributions to ocean mass (ice sheets, glaciers, land water storage) which are propagated to SL fingerprints using the SLE. This work in itself, and a comprehensive documentation of resulting ocean mass trends and discrepancies between estimates at the regional level, would deserve a publication. The authors also derive uncertainties on ocean mass trends which they separate into temporal uncertainty (the amount of uncertainty coming from the natural variability in records), spatial-structural (coming from the fact that the position of sources is not exactly known) and intrinsic (uncertainty in the data itself, the way we measure it for example). This represents a large part of the paper (methods are well documented and the results well presented).

Thank you for these positive comments.

My main concern about this paper is that the representativity (or accuracy) of these uncertainties is not discussed, despite what appears (to me at least) to be inconsistencies across datasets.

Thank you for calling attention to the lack of discussion about the uncertainties representativity. We have incorporated your remarks below, leading to a better discussion of the uncertainties.

I'll try to give a few examples below:

* Section 3.1 and Figure 2 are dedicated to the noise model selection. No information is provided about the goodness of fit of the selected (optimal) noise model. As far as we are told, all models could largely fail at representing the variability in the records (I'm pretty sure this is not the case, but please provide a metric). This could help with the interpretation of Figure 2 where discrepancies between noise models fitted to different datasets are striking: to me this means that the datasets are unable to observe the same processes, even between two GRACE based datasets.

The goodness of the fit was assessed by using the modified Bayesian Information Criterion, as explained in lines 151-153. We have clarified this by adding the following information to the methods section 2.2.2 and the histogram as Figure A2 in the Appendix:

*"Since these criteria are relative values, they can not be compared between different data sets. Thus, we compare the criteria of different noise models for each data set and each grid point separately. To select the best noise model, we compute the relative likelihood of BIC_tp, and select the model with values smaller than 2 (Burnham & Anderson, 2002; Camargo et al., 2020). Note that all noise models reasonably capture the variability of the time series (Figure A2), as their scores are always within a similar range."*

[Figure]

*Figure 1. Histograms of the BIC_tp score for each dataset, used to select the optimal noise models. x-axis shows the BIC score, and y-axis the number of grid points (count). Note that all models have scores within the same range, showing that no model fails in capturing the signal of the observation. However, some models have a slighter lower score, and those will be the ones selected as the optimal noise model. (included in Appendix as Figure A1)*

A synthetic table of datasets resolution/content could be useful here (rather than having the refer to Appendix A).

We have added an extra column to Table 1 in the main text, which now includes the information of the spatial resolution in the dataset (see table below). However, we prefer to keep the more detailed information in Appendix A, for clarity.

* Section 2.2.2 describes the uncertainty propagation rules used in the paper. I'm concerned by two points here:

1. When considering spatial-structural uncertainties the authors scale the fingerprints to 1 mm/yr amplitude to avoid too much spread across an ensemble of only 4 members. How much uncertainty reduction does this scaling provide? This should be documented.

We perform the normalization to isolate the uncertainty due to the spatial pattern of the mass change which is fed into the sea-level equation. If we would take the unscaled standard deviation of the datasets instead, the result would represent more than just uncertainty due to the spatial pattern because each data set has a different magnitude.

We illustrate the difference between normalizing and not normalizing in the figure below (right panel is Figure 4 of the manuscript). This shows that the normalization highlights the differences close to the mass

change source, while without the normalization the differences in the far field are highlighted, as a result of the different global mean values in each dataset.

[Figure]

*Figure 2. Structural uncertainty without the normalization (left) and with normalization (right, figure 5 in the revised manuscript).*

We have rewritten the Methods Section as follows:

"*The second uncertainty we consider is the spatial-structural uncertainty (Figure 1b, right column). Studies that combine a large number of datasets often base the structural uncertainty of an estimate on the standard deviation over the individual datasets in relation to the ensemble mean (Palmer et al., 2021; Cazenave et al., 2018}. However, the small number of samples in our study (4 estimates for each contribution) could lead to unrealistic structural uncertainties when simply based on the standard deviation, as individual outliers could bias the ensemble mean. Instead, we compute the spatial-structural uncertainty by estimating the standard deviation based on the normalized fingerprint for each contribution. First, we use the trend of each contribution to compute sea-level fingerprints normalized to 1 mm/year of global mean SLC.* The aim of the normalization is to isolate the effect that the mass source distribution has on the fingerprint pattern. In other words, it removes the influence that the different central estimates (mean) have on the spatial standard deviation. *We then take the standard deviation across the four (normalized) datasets for each mass source contribution, which leads to four normalized spatial-structural uncertainties reflecting the uncertainty associated with the different spatial resolutions of the datasets.*"

2. Regarding the intrinsic uncertainty propagation, the authors use a no-covariance hypothesis. There are many ways error covariances could affect GRACE/GRACE-FO measurements (instrument ageing, operation mode switches). I understand the scope of the paper is not to revisit GRACE error characterization but this could be mentioned ?

Indeed, instrument ageing and switches on operation mode will influence the signal of GRACE/GRACE-Fo measurements. However, the intrinsic uncertainties discussed here represent the errors in the monthly GRACE gravity field solutions, arising from measurement, processing and aliasing errors. We have added a comment to clarify what the intrinsic uncertainties represent.

These errors will exhibit covariances, which were not included previously, as pointed out by both reviewers. We have therefore modified the intrinsic uncertainty computation to now include error covariance. The intrinsic uncertainty is propagated by perturbing the time series with random noise multiplied by the uncertainty for 1000 times. We then compute the trend for each time series, and use the 95% confidence interval of the distribution as the intrinsic uncertainty. We have adapted the method section accordingly.

https://doi.org/10.5194/esd-2021-80-RC1

*"The final type of uncertainty considered in our assessment is the intrinsic uncertainty, which represents the formal errors and sensitivities in the measurement system and needs to be provided with the observations/models by the data processor/distribution center. The intrinsic uncertainty was only provided with the JPL and IMBIE datasets. For all other cases, the uncertainty budget does not include the intrinsic uncertainty.* **The uncertainties provided with the JPL Mascons represent the scaling and leakage errors from the mascon approach (Wiese et al., 2016), and, over land, are scaled to approximate the formal GRACE uncertainty of Wahr et al. (2006). The latter represent errors in monthly GRACE gravity solutions, encompassing measurement, processing and aliasing errors (Wahr et al., 2006).** *Note that, while the mascons have been corrected for mass changes due to the glacial isostatic adjustment (GIA) with the ICE6G-D model (Peltier et al., 2018), the intrinsic uncertainties of JPL mascons do not represent the uncertainties from the GIA correction, which can be large depending on the region (Reager et al., 2016; Wouters et al., 2019). For example, the choice of the GIA model used for the correction could lead to uncertainties representing 19 percent of the signal in Antarctica, but less than one percent in Greenland (Blazquez et al., 2018). Since estimating GIA uncertainties is in itself an open issue (Caron et al., 2018; Simon and Riva, 2020), we did not attempt to propagate full GIA uncertainties into the fingerprints.* **Since the intrinsic uncertainty represents systematic errors and instrumental noise, which might be serially correlated, we assume that the errors can be approximated by a random walk. We therefore generate an ensemble of 1,000 time series by perturbing the original rate with random normal noise multiplied by the uncertainty time series. We then compute the trend for each ensemble member. We use the half width of the 95% CI as input in the SLE model, to show how the mass associated with the intrinsic uncertainty is distributed over the oceans "**

* It is unclear to me if all datasets are consistent within the estimated uncertainties of if there remains regions where this is not the case ?

We are not sure what the reviewer means with this comment. For a given contribution, the estimated trends based on different datasets agree within their respective uncertainties. Regarding the temporal uncertainties, even when using different noise models (Figure 2), the uncertainties are consistent for each dataset When comparing the different contributions (datasets), then the LWS was much higher than the uncertainties of AIS, GIS and GLA, for the 3 types of uncertainties considered here. But within a contribution (e.g., LWS), the uncertainties of the different datasets (CSR, JPL, WGP, GWB) were consistent. We hope this answers the reviewer's question.

* I have the feeling that the uncertainty estimation presented here is likely a lower bound to the true uncertainty on the ocean mass contribution to SL. Could you comment on that ?

We would actually argue that the uncertainties presented here are an upper bound, although there is no straightforward way of knowing the true uncertainty of ocean mass contribution to sea-level change. Most studies include only one type of uncertainty ( intrinsic, temporal or structural). The objective of our work was to characterize all of these types of uncertainties, and thus we find larger uncertainties than other studies. Nonetheless, we did assume independence of the different types of uncertainty, and did not propagate GIA uncertainties into our fingerprints. Hence, while we do not know the true barystatic uncertainty, and have in general higher estimates than the ones published so far, it is possible that we are still underestimating the true uncertainty on the ocean mass contribution to sea-level change.

We have added a comment about it in the discussion:

*"In this study we assessed the uncertainties related to the barystatic-GRD contribution to regional SLC, in particular the spatial distribution of the uncertainties.* **There is no straightforward way of knowing the true uncertainty of ocean mass contribution to sea-level change. Compared to previous studies (e.g., Horwarth et al., 2021), we tend to find larger uncertainties, thus our approach seems to be conservative. Nonetheless, we did assume independence of the different types of uncertainty, and did not propagate GIA uncertainties into our fingerprints, which could lead to even larger uncertainties.** *Our results highlight that improving the spatial detail of land ice mass loss products, as well as determining more accurate land water storage trends, would lead to better SLC estimates. In addition, our findings can be used to inform projection frameworks. For example, we show that the distribution of ice in the Antarctic Ice*

*Sheet has a significant impact on regional SLC, even in locations far from the ice sheets, such as The Netherlands. This means that, depending on the region of a collapse in the Antarctic Ice Sheet, the sea-level rise projections, which are often based on uniform ice sheet distributions and static fingerprints (e.g., Slangen et al., 2012, Jevrejeva et al, 2019) , may have large regional deviations due to spatial differences in the mass source. Incorporating the insights of uncertainty assessments in sea-level frameworks (as in Larour et al., 2020) should eventually lead to better sea-level projections."*

---

## Author Comment (AC2)

*We thank the reviewer for the positive evaluation and valuable comments. We provide a point-by-point response to each comment, with reviewer's comments in black, and author's responses in purple. The modified manuscript text is shown in quotation marks and italic, with additions in bold.*

**General remarks**

GR1 Intrinsic uncertainties:
The reported uncertainties from the individual data sets are assumed to be uncorrelated. Therefore, the trends are subsequently estimated using ordinary least squares with the uncertainties on the diagonal of the co- variance matrix. I don't think that this is the right approach to estimate the trend uncertainties due to intrinsic errors. For many estimates the uncertainties will be serially correlated, for example due to uncertainties that affect the trend. Like if the uncertainty in a GRACE time series is fully due to GIA uncertainties that only affect the trend, the aforementioned assumption doesn't hold, and the resulting errors are an underestimation. For estimates where the rates and their uncertainties are available, an approximation that often works well is to assume that the rate errors are uncorrelated, which is equal to assuming random walk. Then you can generate an ensemble of time series by perturbing the rate with random normal noise, and then integrate the rates to obtain the time series. A good way to verify the results is to compare the barystatic trend uncertainties in Gt/yr or mm/yr with those reported in the papers where the data sets came from.

*Thank you for pointing this out. This was also noted by Reviewer #1. Indeed, we assumed intrinsic uncertainties to be uncorrelated, but we agree with the reviewers that this assumption may not be valid in this case. We have therefore modified the intrinsic uncertainty computation by perturbing each time series with normal noise for a 1000-member ensemble. We then take the trend for each ensemble member, and compute the 95% confidence interval of the trend distribution. We use the half width of the 95% confidence interval as the rate uncertainty. An example of the new rates are shown in Figure 1 for the GMSL contribution from the AIS and the GIS based on the IMBIE datasets. We note, however, that these uncertainties are still smaller than the ones reported by IMBIE (Shepherd et al., 2018, 2020), as the reported uncertainties include both the intrinsic and temporal components.*

[Figure]

*Figure 1. Mass change rates and improved intrinsic uncertainties. Gray lines on the top panels represent the 1000 ensemble members of the original time series (black) perturbed with random noise and the uncertainty time series (red). Histograms on the lower panels represent the trends of the 1000 ensemble members. We take the 95% confidence interval of the trend distribution, and use the largest difference between the mean and the CI as the rate uncertainty.*

https://doi.org/10.5194/esd-2021-80-RC2

GR2 Trend uncertainties:

I think the paper could use some more discussion on the meaning of the trend uncertainties, because their meaning is not trivial and explaining the meaning of these trend uncertainties is important for data users to correctly apply them. This is a bit of a philosophical point, but the auto-correlated residuals after estimating the trend are not per se due to measurement errors, but they could represent a real signal. An example is the drop in GMSL during the 2010/2011 La Nina event, and the acceleration in ice-sheet mass loss. Let's assume now that someone downloads the regional data as well as some altimetry data of regional sea level. Then the uncertainty in the unexplained residual (Local altimetry – local ocean mass) should not contain the trend uncertainty given here. Altimetry will also see the acceleration and La Nina-like bumps and throughs and the difference probably shows much less interannual variations and thus has a lower trend uncertainty. However, when this user just uses the provided trend uncertainty and adds it in quadrature to some other errors, he or she will overestimate the uncertainties of the just computed difference. Some guidance could help here.

We agree that the residuals could actually represent a real signal, hence the importance of using the noise models to better represent these uncertainties. As noted by the reviewer, depending on the source of this uncertainty, the signal will be present in both the observed mass contributions as well as in altimetry (and also in other components of the sea-level budget). Consequently, summing up the uncertainties of different sources can lead to an overestimation of the uncertainty. We have added a comment about this in the discussion:

*"In this manuscript we investigate the barystatic contribution to regional sea-level trends over 1993-2016 and 2003-2016, focusing on improving the understanding of the uncertainty budget. We show how mass changes of glaciers, land water storage, and the Greenland and Antarctic ice sheets influence regional SLC by computing sea-level fingerprints. We consider three types of uncertainties in our budget: the determination of a linear trend (temporal); the spread around a central estimate as influenced by the distribution of mass change sources (spatial); and the uncertainty from the data/model itself (intrinsic). The uncertainty budget is dominated by the temporal uncertainty, followed by a significant contribution of the spatial-structural uncertainty, while the contribution of the intrinsic uncertainty is relatively small. Regarding the temporal uncertainties associated with the trend, we note that they could partially represent real climatic signals in addition to measurement errors. For example, the variability due to climatic oscillations, such as El Niño Southern Oscillation (ENSO) and the Pacific Decadal Oscillation (PDO), may be reflected in the residuals of the time series, affecting the trend and its temporal uncertainties (Royston et al., 2018). As such climatic events influence not only mass change, but also other drivers of sea-level change (e.g., thermal expansion), caution must be taken when using and comparing these uncertainties with those from other sea level observations."*

GR3 Barystatic sea level:

How do the global-mean time series and uncertainties look like? A simple plot with the global-mean time series from each component and the total might be a nice addition. That could also help verifying the 1993-2016 time series from models: do they show similar trends and variability as GRACE? For TWS, Scanlon et al. (2018) show some discrepancies between models and GRACE for TWS. Might be interesting to see whether estimates from WaterGAP and PCR-GLOBWB now perform better.

We have added a figure of the global-mean time series in the main text as Figure 1, and have included a table with global mean barystatic trends and uncertainties in the appendix (Table A1), as the focus of this study is on regional variations.

[Figure]

*Figure 2. Barystatic contributions to the GMSL time series. Time series are offset by contribution. (included in main text as Figure 1)*

**Table A1.** Table with global mean barystatic sea-level changes and uncertainties from the original global mean timeseries. Note that these numbers may be different compared to the histograms of Figure 7, which represent the spatial average of the regional trend and uncertainty. The difference between the trends is due to the use of noise-models for the regional trend, against an ordinary least-squares fit for the global mean trend.

| | 2003-2016 | | | | | | 1993-2016 | | | | | |
|---|---|---|---|---|---|---|---|---|---|---|---|---|
| | **trend** | ± | $\sigma_{total}$ | $\sigma_{temporall}$ | $\sigma_{spatial}$ | $\sigma_{intrinsic}$ | **trend** | ± | $\sigma_{total}$ | $\sigma_{temporall}$ | $\sigma_{spatial}$ | $\sigma_{intrinsic}$ |
| **AIS** | | | | | | | | | | | | |
| AIS_CSR | 0,32 | ± | 0,09 | 0,03 | 0,09 | | | | | | | |
| AIS_JPL | 0,27 | ± | 0,1 | 0,04 | 0,09 | 0,04 | | | | | | |
| AIS_IMB | 0,37 | ± | 0,13 | 0,05 | 0,09 | 0,07 | 0,19 | ± | 0,15 | 0,04 | 0,14 | 0,03 |
| AIS_UCI | 0,48 | ± | 0,09 | 0,01 | 0,09 | | 0,4 | ± | 0,14 | 0,01 | 0,14 | |
| **GIS** | | | | | | | | | | | | |
| GIS_CSR | 0,72 | ± | 0,32 | 0,03 | 0,31 | | | | | | | |
| GIS_JPL | 0,73 | ± | 0,32 | 0,03 | 0,31 | 0,01 | | | | | | |
| GIS_IMB | 0,53 | ± | 0,32 | 0,03 | 0,31 | 0,07 | 0,36 | ± | 0,12 | 0,03 | 0,11 | 0,03 |
| GIS_UCI | 0,06 | ± | 0,32 | 0,08 | 0,31 | | 0,52 | ± | 0,12 | 0,03 | 0,11 | |
| **GLA** | | | | | | | | | | | | |
| GLA_CSR | 0,68 | ± | 0,16 | 0,06 | 0,15 | | | | | | | |
| GLA_JPL | 0,64 | ± | 0,16 | 0,07 | 0,15 | 0,01 | | | | | | |
| GLA_WGP | 0,58 | ± | 0,15 | 0,03 | 0,15 | | 0,51 | ± | 0,16 | 0,03 | 0,16 | |
| GLA_ZMP | 0,92 | ± | 0,15 | 0,03 | 0,15 | | 0,74 | ± | 0,17 | 0,04 | 0,16 | |
| **LWS** | | | | | | | | | | | | |
| LWS_CSR | 0,09 | ± | 0,14 | 0,12 | 0,06 | | | | | | | |
| LWS_JPL | 0,22 | ± | 0,33 | 0,12 | 0,06 | 0,3 | | | | | | |
| LWS_WGP | 0,2 | ± | 0,12 | 0,1 | 0,06 | | | ± | 0,07 | 0,04 | 0,06 | |
| LWS_GWB | 0,18 | ± | 0,12 | 0,1 | 0,06 | | | ± | 0,07 | 0,04 | 0,06 | |
| **Combination** | | | | | | | | | | | | |
| CSR | 1,81 | ± | 0,39 | 0,14 | 0,36 | | | | | | | |
| JPL | 1,86 | ± | 0,49 | 0,15 | 0,36 | 0,3 | | | | | | |
| IMB+WGP | 1,68 | ± | 0,39 | 0,12 | 0,36 | 0,1 | 1,27 | ± | 0,26 | 0,07 | 0,25 | 0,04 |
| IMB+GWB+ZMP | 2,00 | ± | 0,39 | 0,12 | 0,36 | 0,1 | 1,58 | ± | 0,26 | 0,08 | 0,25 | 0,04 |
| UCI+WGP | 1,32 | ± | 0,38 | 0,13 | 0,36 | | 1,64 | ± | 0,25 | 0,06 | 0,25 | |
| UCI+GWB+ZMP | 1,64 | ± | 0,38 | 0,13 | 0,36 | | 1,95 | ± | 0,26 | 0,06 | 0,25 | |

GR4 The term 'regional barystatic':
After Gregory et al. (2019), barystatic sea level should only refer to global-mean sea-level changes, and not to regional patterns. I'd remove the term 'regional barystatic' and replace it with 'Barystatic sea level and associated GRD patterns' or something like that.

For brevity, we replace 'regional GRD-induced patterns associated with barystatic sea-level change' by 'GRD-induced sea-level change'. We define the term at the end of the introduction:

*"The aim of this work is to provide a comprehensive overview of regional GRD-induced patterns associated with barystatic SLC with a focus on the global and regional uncertainty budget. Throughout this paper, we use 'GRD-induced SLC' when referring to the GRD-induced pattern associated with barystatic SLC."*

GR5 Glaciers and ice sheets:
A nasty problem when working with GRACE data for glaciers is that the GRACE resolution is pretty coarse compared to the size of some glaciers. Therefore, it's hard to separate the mass loss from peripheral glaciers in Greenland and Antarctica from the nearby ice sheets. The same goes for small glaciers in for example Asia, where mass changes from nearby TWS changes leak into the glacier mass change estimates. Did the authors take this into account? A possible way out is the method described in the supporting information of Reager et al. (2016): for the RGI regions where glacier mass loss dominates the GRACE mass change estimates, use GRACE, for the ice sheets, treat the glaciers as part of the ice sheets, and for the other regions, use another dataset (for example Malles & Marzeion, 2021 or Hugonnet et al. 2021) to separate glaciers from TWS.

This is a good point. Up to now, we used the RGI regions to isolate the signals of glaciers from GRACE, and peripheral glaciers to Greenland and Antarctica were included with the ice sheets. However, we had not explicitly considered the LWS leakage into glacier mass change of other regions. Following the suggestion of the reviewer, we have now used the method described in Reager et al. (2016) and Fredekerikse et al. (2019), to separate glaciated areas from LWS: (1) Peripheral glaciers to Greenland and Antarctica are included with the ice sheets mass changes; (2) Regions where glaciers dominate the mass changes are considered 'full' glaciers, that is, the land signal in those regions are purely denoted as glacier mass change. These include the RGI regions of Alaska, Arctic Canada North, Arctica Canada South, Iceland, Svalbard, Russian Arctic Islands and Southern Andes; (3) for the remaining RGI regions, we assume that the mass change is partly due to glacier mass change, and partly due to LWS ('split' glaciers). This method results in the  mask in Figure 3

For the 'split' Glaciers, the glacier mass changes are known to be small and land mass changes are dominated by terrestrial water storage variations. To isolate the glacier to the LWS signal, we use the GRACE glacier estimates of Wouters et al. (2019), which have already been corrected for the hydrological signal. We decided to use the estimates of Wouters et al (2019) instead of the geodetic and glaciological measurements from Zemp et al. (2019) to ensure that the glacier estimates between the 4 datasets used here remain as independent as possible. Another possibility would have been to use the model glacier estimates of Malles & Marzeion (2021), however this is an updated glacier model version of the one incorporated in WaterGap, which would introduce some circularity.

By applying this new method to isolate the glacier and LWS signals in the CSR and JPL mascons, we find small differences in the  trends and (spatial and temporal) uncertainties of the Glaciers and LWS  Figure 4 highlights the differences in the trend and temporal uncertainty of the GRACE datasets before using this method (a-h) and after (i-p).

https://doi.org/10.5194/esd-2021-80-RC2

[Figure]

Figure 3. Barystatic contributions mask.

[Figure]

*Figure 4. Difference between previous and new method of splitting up the signal between glaciers and LWS.* **Panels i-p are included in the manuscript in Figure 4.**

GR6 Used datasets.
The authors have collected a large and diverse set of sources for their barystatic estimates. I've listed below a few other data sets that could be added as well. Since new data sets appear all the time, this list isn't exhaustive and should be seen more as an idea rather than a demand to incorporate them in this manuscript.

Thank you for the suggestions of new datasets. Most of these were not available at the time of preparing the manuscript and could therefore not be included at the time, while others were available but were not suited for the purpose of this study. As mentioned by the reviewer, new datasets become available all the time, so this will always be a moving field. At the moment, four datasets are used for each of the contributions, which encompass a wide range of possibilities giving a robust estimate. We have therefore decided to not expand the number of datasets for now.

**Line-by-line comments**

L10: the trend ranges, do they refer to the 95[th] percentile of the gridded field? I'd think the minimum trend will be lower very close to the ice sheet edges

Yes, they represent the 95th percentile of the ocean grids. The minimum values close to the ice sheet are much lower indeed. For example, from 2003-2016 period the minimum values are around -4.5 mm/yr. We have clarified it in the text that we are citing the 95th percentile of the ocean grids.

L63-L64: "The structural uncertainty is related to the use of different datasets of the same process". The structural and intrinsic uncertainties, are they independent? I can imagine that for example in GRACE, there's an uncertainty in some atmospheric correction, and product A uses estimate A and product B uses estimate B for that process. Then parts of the intrinsic uncertainties also end up in the structural uncertainties.

Indeed, depending on the source of the intrinsic uncertainty it can reflect the structural uncertainty We have added a comment about this in the introduction and also in the methodology (section 2.2.3).

*"Errors in the measurement system, known as intrinsic uncertainties (Palmer et al., 2021), describe the sensitivities of choices within a methodology (Thorne, 2021). The intrinsic uncertainties, also referred as observational (Ablain et al., 2019; Prandi et al., 2021) or parametric (Thorne, 2021), need to be determined during the low-level data processing and are usually provided with higher level (ready-to-use) products. Another class of uncertainties originates from the use of different methodologies to describe the same physical system, known as structural uncertainty (Thorne et al., 2005; Palmer et al., 2021). This can be defined as the spread around a central (ensemble) estimate. The structural uncertainty is related to the use of different datasets of the same process. **Note that, depending on the products used in the processing chain, the intrinsic and structural uncertainties could be partially correlated**."*

*"Whereas the trends are added together linearly, we add the uncertainties in quadrature, assuming they are independent and normally distributed. **We acknowledge that this is an important assumption, as it is possible that the intrinsic uncertainty will be reflected in the temporal and structural uncertainties.** For each contribution, we first combine the different types of uncertainty following Equation (3)."*

L131: More of an idea than a comment: there's a lot of people looking for fingerprints to analyze altimetry data, so there might be quite some interest in complementary geocentric sea-level fingerprints.

Thanks for the suggestion. We will make the geocentric fingerprints available with the dataset on 4TU (https://doi.org/10.4121/16778794, currently reserved).

L166 Average – typo

Corrected

L179 The JPL mascon uncertainties do not represent the uncertainties due to the GIA correction. This uncertainty is actually pretty large, but a bit cumbersome to propagate into the final GRD patterns. It can be done by using uncertainty estimates from for example Caron et al. (2018) and Simon & Riva (2020, I'm sure some of the authors are aware of this study). Propagating the full GIA uncertainties into the fingerprints might be a bit too far-fetched for the current manuscript, but it's a good idea to mention that there's additional uncertainty related to GIA in these GRACE estimates.

Agreed. We have added a comment about this in the manuscript, when describing the intrinsic uncertainties of JPL mascons (Section 2.2.2), and also a comment in the discussion (Section 4).

*"The final type of uncertainty considered in our assessment is the intrinsic uncertainty, which represents the formal errors and sensitivities in the measurement system and needs to be provided with the observations/models by the data processor/distribution center. The intrinsic uncertainty was only provided with the JPL and IMBIE datasets. For all other cases, the uncertainty budget does not include the intrinsic uncertainty. The uncertainties provided with the JPL Mascons represent the scaling and leakage errors from the mascon approach (Wiese et al., 2016), and, over land, are scaled to roughly match the formal GRACE uncertainty of Wahr et al. (2006). The latter represent errors in monthly GRACE gravity solutions, encompassing measurement, processing and aliasing errors (Wahr et al., 2006). Note that, while the mascons have been corrected for mass changes due to the glacial isostatic adjustment (GIA) with the ICE6G-D model (Peltier et al., 2018), the intrinsic uncertainties of JPL mascons do not represent the uncertainties from the GIA correction, which can be large depending on the region (Reager et al., 2016; Wouters et al., 2019). For example, the choice of the GIA model used for the correction could lead to uncertainties representing 19% of the signal in Antarctica, but less than 1% in Greenland (Blazquez et al., 2018). Since estimating GIA uncertainties is in itself an open issue (Caron et al., 2018; Simon and Riva, 2020), we could not propagate full GIA uncertainties into the fingerprints."*

*"Compared to previous studies (e.g., Horwarth et al., 2021), we tend to find larger uncertainties, thus our approach seems to be conservative. Nonetheless, we did assume independence of the different types of uncertainty, and did not propagate GIA uncertainties into our fingerprints, which could lead to even larger uncertainties. ."*

Figure 2 It looks like the glacier mass balance from the CSR and JPL mascons has been estimated by splitting up some of the mascons. This is a bit tricky: for some regions, the mass changes of the whole mascon are dominated by small glaciers, and taking a part of the mascon induces an error. The opposite also happens. I'd recommend to not split mascons into smaller pieces. See also GR5 for a possible way out.

We followed the reviewer's suggestion in GR5 for splitting up the mascons (see Author Response for GR5). We acknowledge that there might be errors introduced to the contributions by splitting up the mascons, but this is a known limitation inherent to GRACE resolution.

L255: Just out of curiosity: does the UCI dataset show any mass gains in East Antarctica?

Yes, the UCI dataset shows mass gains in East Antarctica, but they are very small. For details, the reviewer is referred to Rignot et al. (2019).

L264: as a rule of thumb, the individual mascons from the JPL solution are all independent and agree more-or- less with the spatial resolution of the GRACE measurements. For other mascons, like GSFC and CSR ones that have a much higher resolution, the individual mascons are not fully independent of each other.

Indeed, the JPL mascons are defined on an equal-area of 3x3 degrees, while CSR and GSFC are defined o a 1x1 degree cap. Considering the native resolution of GRACE observations of about 300km half-width at the equator, the JPL mascons will have independent solutions at each mascon centers, with uncorrelated

errors, while the mascons with higher resolution will not be fully independent and is expected to contain spatially correlated errors. We have added a comment about the mascons resolution in section 2.1, when introducing the datasets used.

In this specific line where the reviewer made the comment, we actually wanted to highlight the spatial difference between GRACE observations and the IMBIE and UCI ones, despite the resolution of the mascon solution. Thus, we have rewritten as follows:

*"The regional SLC fingerprints directly reflect the differences in the spatial distribution of the mass change sources of the datasets (Mitrovica et al., 2011). Over the ice sheets, for instance, IMBIE provides one time series for the entire Greenland Ice Sheet, which is subdivided into dynamic and surface mass balance changes, and the Antarctic Ice Sheet is divided into three drainage basins. **GRACE products, on the other hand, have a native resolution of about 300 km at the equator (Tapley et al., 2004).** To account for the uncertainties arising from the differences in location of the mass change between datasets, we first normalize the fingerprints and then combine them into estimates of the spatial-structural uncertainty (Figure 4)."*

Figure 4: This is a very interesting figure! I discovered a lot of intriguing phenomena when looking at it.

Thanks

Figure 5: also related to GR1. If you check the uncertainties listed in Table 1 from the IMBIE Antarctic paper, the reported uncertainties, which are about 50 Gt yr-1 for 5-year periods, seem to be much higher than reported here. This is probably related to the assumption of uncorrelated uncertainties. Using the rates+uncertainties procedure from GR1 might solve this difference.

Using the 'ensemble perturbation' method, as suggested by the reviewer in GR1, we obtained higher intrinsic uncertainties than before, but they are still smaller than the ones reported in the IMBIE reports. This is because their reported uncertainties represent not only the intrinsic uncertainty, but also the temporal uncertainties. We have added a comment about this in the manuscript:

*" The IMBIE datasets (Figure 6e,f) show larger intrinsic uncertainty than the ice sheet uncertainties from JPL (Figure 6c,d), which is expected as the IMBIE time series is an ensemble of several datasets and methods. **Note that these uncertainties are smaller than those reported in the IMBIE studies (Shepherd et al., 2018, 2020), because the reported uncertainties represent not only the use of the uncertainty time series (intrinsic), but also the errors due to the linear fit of the trend (temporal uncertainties)."***

L375: Check the paper from Humphrey and Gudmundsson (2019), who provide some centennial estimates of TWS changes.

Thank you for the literature suggestion. Humphery and Gudmudsson (2019) discuss deseasonalized and detrended time series, and their work focuses on the validation of their variability dataset instead of the discussion of the results, so it actually can't be used in our discussion of trends. However your suggestion did lead to another work by the same authors which was useful for the discussion here and is added to Section 4.

*"The main source of uncertainty in the barystatic-GRD SLC is the temporal uncertainty from the land water storage (LWS) contribution. This is likely related to the (climate-driven) natural variability of LWS (Vishwakarma et al., 2021; Hamlington et al., 2017; Nerem et al., 2018), which is mainly driven by seasonal and interannual cycles (Cáceres et al., 2020). A method to deal with the LWS natural variability would be to use different metrics than linear trends (Vishwakarma et al., 2021), such as the use time varying trends based on a state space model (Frederikse et al., 2016; Vishwakarma et al., 2021). However, we choose to use linear trends in this study for sake of accuracy, reproducibility and discussion. It has also been suggested that a more appropriate way of computing a meaningful linear trend from LWS is to incorporate this variability in the analysis (Vishwakarma et al., 2021), as we did by including the seasonal components in the functional model. Nonetheless, the LWS uncertainties related to the trend were still*

*very high, suggesting that a period of 25 years (1993-2016) might still be too short to solve the low frequency natural variability of LWS, particularly on (multi)-decadal timescales.* **Indeed, Humphrey et al. (2017) showed that removing the short-term climate-driven variability of the LWS signal yields in a more robust long-term (>10 years) trend, with reduced uncertainties**"

L381: Antarctica Ice Sheet typo

Corrected

L429 Individuals typo

Corrected

---

## Author Response (AR1)

We thank the reviewer for the positive evaluation and valuable comments. We provide a point-by-point response to each comment, with reviewer's comments in black, and author's responses in purple. The modified manuscript text is shown in quotation marks and italic, and we indicate the line numbers of the revised manuscript.

This work compares different estimates of ocean mass contributions to sea level rise. These contributions are themselves derived from different estimates of contributions to ocean mass (ice sheets, glaciers, land water storage) which are propagated to SL fingerprints using the SLE. This work in itself, and a comprehensive documentation of resulting ocean mass trends and discrepancies between estimates at the regional level, would deserve a publication. The authors also derive uncertainties on ocean mass trends which they separate into temporal uncertainty (the amount of uncertainty coming from the natural variability in records), spatial-structural (coming from the fact that the position of sources is not exactly known) and intrinsic (uncertainty in the data itself, the way we measure it for example). This represents a large part of the paper (methods are well documented and the results well presented).

Thank you for these positive comments.

My main concern about this paper is that the representativity (or accuracy) of these uncertainties is not discussed, despite what appears (to me at least) to be inconsistencies across datasets.

Thank you for calling attention to the lack of discussion about the uncertainties representativity. We have incorporated your remarks below, leading to a better discussion of the uncertainties.

I'll try to give a few examples below:

* Section 3.1 and Figure 2 are dedicated to the noise model selection. No information is provided about the goodness of fit of the selected (optimal) noise model. As far as we are told, all models could largely fail at representing the variability in the records (I'm pretty sure this is not the case, but please provide a metric). This could help with the interpretation of Figure 2 where discrepancies between noise models fitted to different datasets are striking: to me this means that the datasets are unable to observe the same processes, even between two GRACE based datasets.

The goodness of the fit was assessed by using the modified Bayesian Information Criterion, as explained in lines 172-174. We have clarified this by adding the following information to the methods section 2.2.2   (Lines 174-178) and the histogram as Figure A2  in the Appendix:

*"Since these criteria are relative values, they can not be compared between different data sets. Thus, we compare the criteria of different noise models for each data set and each grid point separately.  The best noise model is the one that minimizes these criteria. Since these criteria are relative values, they can not be compared between different data sets. Thus, we compare the criteria of different noise models for each data set and each grid point separately. To select the best noise model, we compute the relative likelihood of the $BIC_{tp}$, and select the model with values smaller than 2 (Burnham and Anderson, 2002; Camargo et al., 2020). Note that all noise models reasonably capture the variability of the time series (Figure A2), as their scores are always within a similar range. "*

[Figure]

*Figure 1. Histograms of the BIC_tp score for each dataset, used to select the optimal noise models. x-axis shows the BIC score, and y-axis the number of grid points (count). Note that all models have scores within the same range, showing that no model fails in capturing the signal of the observation. However, some models have a slighter lower score, and those will be the ones selected as the optimal noise model. (included in Appendix as Figure A1)*

A synthetic table of datasets resolution/content could be useful here (rather than having the refer to Appendix A).

We have added an extra column to Table 1 in the main text, which now includes the information of the spatial resolution in the dataset. However, we prefer to keep the more detailed information in Appendix A, for clarity.

* Section 2.2.2 describes the uncertainty propagation rules used in the paper. I'm concerned by two points here:

1. When considering spatial-structural uncertainties the authors scale the fingerprints to 1 mm/yr amplitude to avoid too much spread across an ensemble of only 4 members. How much uncertainty reduction does this scaling provide? This should be documented.

We perform the normalization to isolate the uncertainty due to the spatial pattern of the mass change which is fed into the sea-level equation. If we would take the unscaled standard deviation of the datasets

instead, the result would represent more than just uncertainty due to the spatial pattern because each data set has a different magnitude.

We illustrate the difference between normalizing and not normalizing in the figure below (right panel is Figure 4 of the manuscript). This shows that the normalization highlights the differences close to the mass change source, while without the normalization the differences in the far field are highlighted, as a result of the different global mean values in each dataset.

[Figure]

*Figure 2. Structural uncertainty without the normalization (left) and with normalization (right, figure 5 in the revised manuscript).*

We have adjusted the Methods Section (Lines 181-187) to clarify the aim of the normalization, as follows:

*"To isolate the effect that the spatial distribution of the terrestrial mass change has on the fingerprints, we compute the spatial-structural uncertainty by estimating the standard deviation for each contribution based on normalized fingerprints. The latter means that the sum of the regional SLC for each contribution is equal to 1 mm.year$^{-1}$ of SLC. By using normalized fingerprints we remove the weight that the different central estimates (mean) have on the spatial standard deviation. We then take the standard deviation across the four normalized datasets for each mass source contribution, obtaining four normalized spatial-structural uncertainties, which reflects the uncertainty associated with the different spatial resolutions and location of mass change of the datasets."*

2. Regarding the intrinsic uncertainty propagation, the authors use a no-covariance hypothesis. There are many ways error covariances could affect GRACE/GRACE-FO measurements (instrument ageing, operation mode switches). I understand the scope of the paper is not to revisit GRACE error characterization but this could be mentioned ?

Indeed, instrument ageing and switches on operation mode will influence the signal of GRACE/GRACE-Fo measurements. However, the intrinsic uncertainties discussed here represent the errors in the monthly GRACE gravity field solutions, arising from measurement, processing and aliasing errors. We have added a comment to clarify what the intrinsic uncertainties represent.

These errors will exhibit covariances, which were not included previously, as pointed out by both reviewers. We have therefore modified the intrinsic uncertainty computation to now include error covariance. The intrinsic uncertainty is propagated by perturbing the time series with random noise multiplied by the

uncertainty for 1000 times. We then compute the trend for each time series, and use the 95% confidence interval of the distribution as the intrinsic uncertainty.

We added a comment about the GRACE errors and adjusted the intrinsic computation methods as follows (Lines 200-216):

*"The final type of uncertainty considered in our assessment is the intrinsic uncertainty, which represents the formal errors and sensitivities in the measurement system and needs to be provided with the observations/models by the data processor/distribution centre. The intrinsic uncertainty was only provided with the JPL and IMBIE datasets. For all other datasets, our uncertainty budget does not include the intrinsic uncertainty. The uncertainties provided with the JPL Mascons represent the scaling and leakage errors from the mascon approach (Wiese et al., 2016), and, over land, are scaled to roughly match the formal GRACE uncertainty of Wahr et al. (2006). The latter represent errors in monthly GRACE gravity solutions, encompassing measurement, processing and aliasing errors (Wahr et al., 2006). While the mascons have been corrected for mass changes due to glacial isostatic adjustment (GIA) with the ICE6G-D model (Peltier et al., 2018), the intrinsic uncertainties of the JPL mascons do not represent the uncertainties from the GIA correction, which can be large depending on the region (Reager et al., 2016; Wouters et al., 2019). For example, the choice of the GIA model used for the correction could lead to uncertainties representing up to 19% of the signal in Antarctica, but less than 1% in Greenland (Blazquez et al., 2018). Given that estimating GIA uncertainties is in itself an open issue (Caron et al., 2018; Simon and Riva, 2020), we could not propagate full GIA uncertainties into the fingerprints. Since the intrinsic uncertainty represents systematic errors and instrumental noise, which might be serially correlated, we assume that the errors can be approximated by a random walk. We therefore generate an ensemble of 1,000 time series by perturbing the original rate with random normal noise multiplied by the uncertainty time series. We then compute the trend for each ensemble member. We use half of the width of the 95% CI as input in the SLE model to show how the mass associated with the intrinsic uncertainty is distributed over the oceans."*

\* It is unclear to me if all datasets are consistent within the estimated uncertainties of if there remains regions where this is not the case ?

We are not sure what the reviewer means with this comment. For a given contribution, the estimated trends based on different datasets agree within their respective uncertainties. Regarding the temporal uncertainties, even when using different noise models (Figure 2), the uncertainties are consistent for each dataset When comparing the different contributions (datasets), then the LWS was much higher than the uncertainties of AIS, GIS and GLA, for the 3 types of uncertainties considered here. But within a contribution (e.g., LWS), the uncertainties of the different datasets (CSR, JPL, WGP, GWB) were consistent. We hope this answers the reviewer's question.

\* I have the feeling that the uncertainty estimation presented here is likely a lower bound to the true uncertainty on the ocean mass contribution to SL. Could you comment on that ?

We would actually argue that the uncertainties presented here are an upper bound, although there is no straightforward way of knowing the true uncertainty of ocean mass contribution to sea-level change. Most studies include only one type of uncertainty ( intrinsic, temporal or structural). The objective of our work

was to characterize all of these types of uncertainties, and thus we find larger uncertainties than other studies. Nonetheless, we did assume independence of the different types of uncertainty, and did not propagate GIA uncertainties into our fingerprints. Hence, while we do not know the true barystatic uncertainty, and have in general higher estimates than the ones published so far, it is possible that we are still underestimating the true uncertainty on the ocean mass contribution to sea-level change.

We have added a comment about it in the discussion (Lines 430-432):

*"The true uncertainty of ocean mass contribution to sea-level change is difficult to determine. Our approach of quantifying this uncertainty is to some extent conservative, as it results in larger uncertainties than in previous studies (e.g., Horwath et al. (2021)). Nonetheless, we did assume independence of the different types of uncertainty, and did not propagate GIA uncertainties into our fingerprints, which could lead to even larger uncertainties "*

https://doi.org/10.5194/esd-2021-80-RC1

Review of 'Trends and Uncertainties of Regional Barystatic Sea-level Change in the Satellite Altimetry Era'

In this manuscript, the authors have collected a large dataset of the sources of barystatic sea-level changes, computed the associated GRD fingerprints, and discuss the sources of uncertainty in the resulting GRD pattern. I think this dataset will be very useful for many applications in sea-level and climate research, and I'm convinced this work is a worthy contribution to the literature on this topic.

Nevertheless, I have a few technical and philosophical remarks and questions. I start with some general remarks, followed by line-by-line comments.

Thomas Frederikse

We thank the reviewer for the positive evaluation and valuable comments. We provide a point-by-point response to each comment, with reviewer's comments in black, and author's responses in purple. The modified manuscript text is shown in quotation marks and italic, and we indicate the line numbers of the revised manuscript.

**General remarks**

GR1 Intrinsic uncertainties:
The reported uncertainties from the individual data sets are assumed to be uncorrelated. Therefore, the trends are subsequently estimated using ordinary least squares with the uncertainties on the diagonal of the co- variance matrix. I don't think that this is the right approach to estimate the trend uncertainties due to intrinsic errors. For many estimates the uncertainties will be serially correlated, for example due to uncertainties that affect the trend. Like if the uncertainty in a GRACE time series is fully due to GIA uncertainties that only affect the trend, the aforementioned assumption doesn't hold, and the resulting errors are an underestimation. For estimates where the rates and their uncertainties are available, an approximation that often works well is to assume that the rate errors are uncorrelated, which is equal to assuming random walk. Then you can generate an ensemble of time series by perturbing the rate with random normal noise, and then integrate the rates to obtain the time series. A good way to verify the results is to compare the barystatic trend uncertainties in Gt/yr or mm/yr with those reported in the papers where the data sets came from.

Thank you for pointing this out. This was also noted by Reviewer #1. Indeed, we assumed intrinsic uncertainties to be uncorrelated, but we agree with the reviewers that this assumption may not be valid in this case. We have therefore modified the intrinsic uncertainty computation by perturbing each time series with normal noise for a 1000-member ensemble. We then take the trend for each ensemble member, and compute the 95% confidence interval of the trend distribution. We use the half width of the 95% confidence interval as the rate uncertainty. An example of the new rates are shown in Figure 1 for the GMSL contribution from the AIS and the GIS based on the IMBIE datasets. We note, however, that these uncertainties are still smaller than the ones reported by IMBIE (Shepherd et al., 2018, 2020), as the reported uncertainties include both the intrinsic and temporal components.

https://doi.org/10.5194/esd-2021-80-RC2

[Figure]

*Figure 1. Mass change rates and improved intrinsic uncertainties. Gray lines on the top panels represent the 1000 ensemble members of the original time series (black) perturbed with random noise and the uncertainty time series (red). Histograms on the lower panels represent the trends of the 1000 ensemble members. We take the 95% confidence interval of the trend distribution, and use the largest difference between the mean and the CI as the rate uncertainty.*

We adjusted the intrinsic computation methods as follows (Lines 200-216):

*"The final type of uncertainty considered in our assessment is the intrinsic uncertainty, which represents the formal errors and sensitivities in the measurement system and needs to be provided with the observations/models by the data processor/distribution centre. The intrinsic uncertainty was only provided with the JPL and IMBIE datasets. For all other datasets, our uncertainty budget does not include the intrinsic uncertainty. The uncertainties provided with the JPL Mascons represent the scaling and leakage errors from the mascon approach (Wiese et al., 2016), and, over land, are scaled to roughly match the formal GRACE uncertainty of Wahr et al. (2006). The latter represent errors in monthly GRACE gravity solutions, encompassing measurement, processing and aliasing errors (Wahr et al., 2006). While the mascons have been corrected for mass changes due to glacial isostatic adjustment (GIA) with the ICE6G-D model (Peltier et al., 2018), the intrinsic uncertainties of the JPL mascons do not represent the uncertainties from the GIA correction, which can be large depending on the region (Reager et al., 2016; Wouters et al., 2019). For example, the choice of the GIA model used for the correction could lead to uncertainties representing up to 19% of the signal in Antarctica, but less than 1% in Greenland (Blazquez et al., 2018). Given that estimating GIA uncertainties is in itself an open issue (Caron et al., 2018; Simon and Riva, 2020), we could not propagate*

https://doi.org/10.5194/esd-2021-80-RC2

*full GIA uncertainties into the fingerprints. Since the intrinsic uncertainty represents systematic errors and instrumental noise, which might be serially correlated, we assume that the errors can be approximated by a random walk. We therefore generate an ensemble of 1,000 time series by perturbing the original rate with random normal noise multiplied by the uncertainty time series. We then compute the trend for each ensemble member. We use half of the width of the 95% CI as input in the SLE model to show how the mass associated with the intrinsic uncertainty is distributed over the oceans. "*

GR2 Trend uncertainties:

I think the paper could use some more discussion on the meaning of the trend uncertainties, because their meaning is not trivial and explaining the meaning of these trend uncertainties is important for data users to correctly apply them. This is a bit of a philosophical point, but the auto-correlated residuals after estimating the trend are not per se due to measurement errors, but they could represent a real signal. An example is the drop in GMSL during the 2010/2011 La Nina event, and the acceleration in ice-sheet mass loss. Let's assume now that someone downloads the regional data as well as some altimetry data of regional sea level. Then the uncertainty in the unexplained residual (Local altimetry – local ocean mass) should not contain the trend uncertainty given here. Altimetry will also see the acceleration and La Nina-like bumps and throughs and the difference probably shows much less interannual variations and thus has a lower trend uncertainty. However, when this user just uses the provided trend uncertainty and adds it in quadrature to some other errors, he or she will overestimate the uncertainties of the just computed difference. Some guidance could help here.

We agree that the residuals could actually represent a real signal, hence the importance of using the noise models to better represent these uncertainties. As noted by the reviewer, depending on the source of this uncertainty, the signal will be present in both the observed mass contributions as well as in altimetry (and also in other components of the sea-level budget). Consequently, summing up the uncertainties of different sources can lead to an overestimation of the uncertainty. We have added a comment about this in the discussion (Lines 387-392):

*"The temporal uncertainties associated with the trend may represent real climatic signals, and not only measurement errors. For example, the variability due to climatic oscillations, such as El Niño Southern Oscillation (ENSO) and the Pacific Decadal Oscillation (PDO), may be reflected in the residuals of the time series, affecting the trend and its temporal uncertainties (Royston et al., 2018). As such climatic events influence not only mass change, but also other drivers of sea-level change (e.g., thermal expansion), caution must be taken when using and comparing these uncertainties with those from other sea level contributors."*

GR3 Barystatic sea level:

How do the global-mean time series and uncertainties look like? A simple plot with the global-mean time series from each component and the total might be a nice addition. That could also help verifying the 1993-2016 time series from models: do they show similar trends and variability as GRACE? For TWS, Scanlon et al. (2018) show some discrepancies between models and GRACE for TWS. Might be interesting to see whether estimates from WaterGAP and PCR-GLOBWB now perform better.

https://doi.org/10.5194/esd-2021-80-RC2

We have added a figure of the global-mean time series in the main text (Figure 1 on revised manuscript), and have included a table with global mean barystatic trends and uncertainties in the appendix (Table A1 on revised manuscript), as the focus of this study is on regional variations.

GR4 The term 'regional barystatic':
After Gregory et al. (2019), barystatic sea level should only refer to global-mean sea-level changes, and not to regional patterns. I'd remove the term 'regional barystatic' and replace it with 'Barystatic sea level and associated GRD patterns' or something like that.

For brevity, we replace 'regional GRD-induced patterns associated with barystatic sea-level change' by 'GRD-induced sea-level change'. We define the term at the end of the introduction (Lines 76-77):

*"Throughout this paper, we use 'GRD-induced SLC' when referring to the GRD-induced regional pattern associated with barystatic SLC."*

GR5 Glaciers and ice sheets:
A nasty problem when working with GRACE data for glaciers is that the GRACE resolution is pretty coarse compared to the size of some glaciers. Therefore, it's hard to separate the mass loss from peripheral glaciers in Greenland and Antarctica from the nearby ice sheets. The same goes for small glaciers in for example Asia, where mass changes from nearby TWS changes leak into the glacier mass change estimates. Did the authors take this into account? A possible way out is the method described in the supporting information of Reager et al. (2016): for the RGI regions where glacier mass loss dominates the GRACE mass change estimates, use GRACE, for the ice sheets, treat the glaciers as part of the ice sheets, and for the other regions, use another dataset (for example Malles & Marzeion, 2021 or Hugonnet et al. 2021) to separate glaciers from TWS.

This is a good point. Up to now, we used the RGI regions to isolate the signals of glaciers from GRACE, and peripheral glaciers to Greenland and Antarctica were included with the ice sheets. However, we had not explicitly considered the LWS leakage into glacier mass change of other regions. Following the suggestion of the reviewer, we have added the LWS-GLA correction, using the glacier estimates of Hugonnet et al (2021), which are not affected by LWS variations, since it is based on satellite and airborne elevation measurements. We decided to use the estimates of Hugonnet et al (2021) instead of the geodetic and glaciological measurements from Zemp et al. (2019) to ensure that the glacier estimates between the 4 datasets used here remain as independent as possible. Another possibility would have been to use the model glacier estimates of Malles & Marzeion (2021), however this is an updated glacier model version of the one incorporated in WaterGap, which would introduce some circularity.

We adjusted the methods section accordingly, as follows (Lines 106-118):

*"Considering the size of glaciers, the resolution of the GRACE signal is not high enough to (i) separate the peripheral glaciers from the ice sheets, and (ii) to separate the signal of glaciers and LWS in regions with small glacier coverage and large LWS contribution. Thus, to isolate the glaciers signals from the mascons we follow the method described in Reager et al. (2016) and Frederikse et al. (2019): (1) peripheral glaciers to Greenland and Antarctica are included with the ice sheets mass changes; (2) regions where glaciers*

https://doi.org/10.5194/esd-2021-80-RC2

*dominate the mass changes are considered 'full' glaciers, that is, the land signals in those regions are purely denoted as glacier mass change. These include the RGI regions of Alaska, Arctic Canada North, Arctic Canada South, Iceland, Svalbard, Russian Arctic Islands and Southern Andes; (3) for the remaining glaciated regions, we assume that the mass change is partly due to glacier mass change, and partly due to LWS ('split' glaciers). In these regions the glacier mass changes are known to be small and mass changes are dominated by LWS. We use the glacier estimates of Hugonnet et al. (2021), which are based on satellite and airborne elevation datasets as our glacier estimates in these regions. Unlike gravimetry observations, the estimates of Hugonnet et al. (2021) do not include the hydrological 'contamination'. To isolate the glacier from the LWS signal, we subtract the corrected glacier estimates from the total mass change in the mascons. The remaining signal is then added to the LWS contribution."*

GR6 Used datasets.
The authors have collected a large and diverse set of sources for their barystatic estimates. I've listed below a few other data sets that could be added as well. Since new data sets appear all the time, this list isn't exhaustive and should be seen more as an idea rather than a demand to incorporate them in this manuscript.

Thank you for the suggestions of new datasets. Most of these were not available at the time of preparing the manuscript and could therefore not be included at the time, while others were available but were not suited for the purpose of this study. As mentioned by the reviewer, new datasets become available all the time, so this will always be a moving field. At the moment, four datasets are used for each of the contributions, which encompass a wide range of possibilities giving a robust estimate. We have therefore decided to not expand the number of datasets for now.

**Line-by-line comments**

L10: the trend ranges, do they refer to the 95th percentile of the gridded field? I'd think the minimum trend will be lower very close to the ice sheet edges

These ranges refer to the 5th to 95th percentile range across all grid cells. The minimum values close to the ice sheet are much lower indeed. For example, from 2003-2016 period the minimum values are around -4.5 mm/yr. We have clarified it in the abstract (Line 12), and in the main text (Lines 330-332):

*"To illustrate the distribution of the regional trends and uncertainties around the world, we report the 5th to 95th percentile range across all ocean grid cells (Figure 7, histograms below the maps), and refer to it as the 90%-range of the field."*

L63-L64: "The structural uncertainty is related to the use of different datasets of the same process". The structural and intrinsic uncertainties, are they independent? I can imagine that for example in GRACE, there's an uncertainty in some atmospheric correction, and product A uses estimate A and product B uses estimate B for that process. Then parts of the intrinsic uncertainties also end up in the structural uncertainties.

https://doi.org/10.5194/esd-2021-80-RC2

Indeed, depending on the source of the intrinsic uncertainty it can reflect the structural uncertainty We have added a comment about this in the introduction (Lines 67-68) and also in the methodology (Lines 222-224):

"Note that, if different datasets use the same product for corrections, calibrations and/or validation, the intrinsic and structural uncertainties could be partially correlated."

*"Whereas the trends are added together linearly, we add the uncertainties in quadrature, assuming they are independent and normally distributed. We acknowledge that this is an important assumption, as it is possible that the intrinsic uncertainty will be reflected in the temporal and structural uncertainties. "*

L131: More of an idea than a comment: there's a lot of people looking for fingerprints to analyze altimetry data, so there might be quite some interest in complementary geocentric sea-level fingerprints.

Thanks for the suggestion. We will make the geocentric fingerprints available with the dataset on 4TU (https://doi.org/10.4121/16778794, currently reserved).

L166 Average – typo

Corrected

L179 The JPL mascon uncertainties do not represent the uncertainties due to the GIA correction. This uncertainty is actually pretty large, but a bit cumbersome to propagate into the final GRD patterns. It can be done by using uncertainty estimates from for example Caron et al. (2018) and Simon & Riva (2020, I'm sure some of the authors are aware of this study). Propagating the full GIA uncertainties into the fingerprints might be a bit too far-fetched for the current manuscript, but it's a good idea to mention that there's additional uncertainty related to GIA in these GRACE estimates.

Agreed. We have added a comment about this in the manuscript, when describing the intrinsic uncertainties of JPL mascons (Lines 206-212), and also a comment in the discussion (Lines 431-433).

*"While the mascons have been corrected for mass changes due to glacial isostatic adjustment (GIA) with the ICE6G-D model (Peltier et al., 2018), the intrinsic uncertainties of the JPL mascons do not represent the uncertainties from the GIA correction, which can be large depending on the region (Reager et al., 2016; Wouters et al., 2019). For example, the choice of the GIA model used for the correction could lead to uncertainties representing up to 19% of the signal in Antarctica, but less than 1% in Greenland (Blazquez et al., 2018). Given that estimating GIA uncertainties is in itself an open issue (Caron et al., 2018; Simon and Riva, 2020), we could not propagate full GIA uncertainties into the fingerprints."*

*"Our approach of quantifying this uncertainty is to some extent conservative, as it results in larger uncertainties than in previous studies (e.g., Horwath et al. (2021)). Nonetheless, we did assume independence of the different types of uncertainty, and did not propagate GIA uncertainties into our fingerprints, which could lead to even larger uncertainties "*

Figure 2 It looks like the glacier mass balance from the CSR and JPL mascons has been estimated by splitting up some of the mascons. This is a bit tricky: for some regions, the mass changes of the whole mascon are dominated by small glaciers, and taking a part of the mascon induces an error. The opposite also happens. I'd recommend to not split mascons into smaller pieces. See also GR5 for a possible way out.

We followed the reviewer's suggestion in GR5 for splitting up the mascons (see Author Response for GR5). We acknowledge that there might be errors introduced to the contributions by splitting up the mascons, but this is a known limitation inherent to GRACE resolution.

L255: Just out of curiosity: does the UCI dataset show any mass gains in East Antarctica?

Yes, the UCI dataset shows mass gains in East Antarctica, but they are very small. For details, the reviewer is referred to Rignot et al. (2019).

L264: as a rule of thumb, the individual mascons from the JPL solution are all independent and agree more-or- less with the spatial resolution of the GRACE measurements. For other mascons, like GSFC and CSR ones that have a much higher resolution, the individual mascons are not fully independent of each other.

Indeed, the JPL mascons are defined on an equal-area of 3x3 degrees, while CSR and GSFC are defined o a 1x1 degree cap. Considering the native resolution of GRACE observations of about 300km half-width at the equator, the JPL mascons will have independent solutions at each mascon centers, with uncorrelated errors, while the mascons with higher resolution will not be fully independent and is expected to contain spatially correlated errors. We have added a comment about the mascons resolution in section 2.1 (Lines 95-102), when introducing the datasets used.

*"We use GRACE mass concentrations (mascons) over land as estimates of changes in AIS, GIS, glaciers and LWS. To avoid methodological biases, we use mascon solutions from two different processing centres: RL06 from Center for Spatial Resarch (CSR) (Save et al., 2016; Save, 2020) and RL06 v02 from Jet Propulsion Laboratory (JPL) (Watkins et al., 2015; Wiese et al., 2019) (Table 1). JPL and CSR mascons are provided on a $0.5°$ and $0.25°$ lon-lat grid, respectively, but they actually are resampled from the native $3°x3°$ and $1°x1°$ equal-area grids (Save et al., 2016; Watkins et al., 2015). Considering the native resolution of GRACE observations of about 300km at the equator (Tapley et al., 2004), the JPL mascons should have independent solutions at each mascon centres, with uncorrelated errors, while the CSR mascons are not fully independent of each other and are expected to contain spatially correlated errors."*

In this specific line where the reviewer made the comment, we actually wanted to highlight the spatial difference between GRACE observations and the IMBIE and UCI ones, despite the resolution of the mascon solution. Thus, we have rewritten as follows (Lines 296-298):

*"Over the ice sheets, for instance, IMBIE provides one time series for the entire Greenland Ice Sheet, which is subdivided into dynamic and surface mass balance changes, and the Antarctic Ice Sheet is divided into three drainage basins. GRACE products, on the other hand, have a native resolution of about 300-km at the equator (Tapley et al., 2004)."*

https://doi.org/10.5194/esd-2021-80-RC2

Figure 4: This is a very interesting figure! I discovered a lot of intriguing phenomena when looking at it.

Thanks

Figure 5: also related to GR1. If you check the uncertainties listed in Table 1 from the IMBIE Antarctic paper, the reported uncertainties, which are about 50 Gt yr-1 for 5-year periods, seem to be much higher than reported here. This is probably related to the assumption of uncorrelated uncertainties. Using the rates+uncertainties procedure from GR1 might solve this difference.

Using the 'ensemble perturbation' method, as suggested by the reviewer in GR1, we obtained higher intrinsic uncertainties than before, but they are still smaller than the ones reported in the IMBIE reports. This is because their reported uncertainties represent not only the intrinsic uncertainty, but also the temporal uncertainties. We have added a comment about this in the manuscript (Lines 317-319):

*"Note that these uncertainties are smaller than those originally reported in the IMBIE studies (Shepherd et al., 2018, 2020), which include not only intrinsic, ut also structural and temporal uncertainties."*

L375: Check the paper from Humphrey and Gudmundsson (2019), who provide some centennial estimates of TWS changes.

Thank you for the literature suggestion. Humphery and Gudmudsson (2019) discuss deseasonalized and detrended time series, and their work focuses on the validation of their variability dataset instead of the discussion of the results, so it actually can't be used in our discussion of trends. However your suggestion did lead to another work by the same authors which was useful for the discussion here and is added to Section 4 (Lines 426-428).

*"Nonetheless, the LWS uncertainties related to the trend are still very high, suggesting that a period of 25 years (1993-2016) might still be too short to solve the low frequency natural variability of LWS, particularly on (multi)-decadal timescales. Indeed, Humphrey et al. (2017) showed that removing the short-term climate-driven variability of the LWS signal yields in a more robust long-term (>10 years) trend, with reduced uncertainties."*

L381: Antarctica Ice Sheet typo

Corrected

L429 Individuals typo

Corrected

https://doi.org/10.5194/esd-2021-80-RC2